# A rare variant analysis framework using public genotype summary counts to prioritize disease-predisposition genes

Wenan Chen [1,7]✉, Shuoguo Wang[2,6], Saima Sultana Tithi[3], David W. Ellison[4], Daniel J. Schaid[5] & Gang Wu [1,4,7]✉

Sequencing cases without matched healthy controls hinders prioritization of germline disease-predisposition genes. To circumvent this problem, genotype summary counts from public data sets can serve as controls. However, systematic inflation and false positives can arise if confounding factors are not controlled. We propose a framework, consistent summary counts based rare variant burden test (CoCoRV), to address these challenges. CoCoRV implements consistent variant quality control and filtering, ethnicity-stratified rare variant association test, accurate estimation of inflation factors, powerful FDR control, and detection of rare variant pairs in high linkage disequilibrium. When we applied CoCoRV to pediatric cancer cohorts, the top genes identified were cancer-predisposition genes. We also applied CoCoRV to identify disease-predisposition genes in adult brain tumors and amyotrophic lateral sclerosis. Given that potential confounding factors were well controlled after applying the framework, CoCoRV provides a cost-effective solution to prioritizing disease-risk genes enriched with rare pathogenic variants.

[1] Center for Applied Bioinformatics, St. Jude Children's Research Hospital, Memphis, TN, USA. [2] Department of Computational Biology, St. Jude Children's Research Hospital, Memphis, TN, USA. [3] Department of Cell & Molecular Biology, St. Jude Children's Research Hospital, Memphis, TN, USA. [4] Department of Pathology, St. Jude Children's Research Hospital, Memphis, TN, USA. [5] Department of Quantitative Health Sciences, Mayo Clinic, Rochester, MN, USA. [6] Present address: 150 Second Street, Cambridge, MA, USA. [7] These authors jointly supervised this work: Wenan Chen, Gang Wu. ✉email: wenan.chen@stjude.org; gang.wu@stjude.org

To identify genetic variants, especially rare variants, that are associated with various human diseases, scientists have generated whole-exome sequencing (WES) and whole-genome sequencing (WGS) data, such as the Genome Aggregation Database (gnomAD)[1], the Trans-Omics for Precision Medicine (TOPMed) program[2], and the 100,000 Genomes Project[3]. Detecting causative rare variants typically requires a much larger sample size. Such a study should include at least thousands of cases and matched controls to ensure statistical power. However, most sequencing studies focused on a specific disease or trait include relatively few cases and very few (if any) matched healthy controls. Although many large cancer genomics studies have characterized the landscape of somatic mutations by sequencing germline and somatic samples from each patient, these studies do not include independent controls because germline samples from the same individuals are used as paired normal controls. Therefore, most cancer genomics sequencing studies cannot be used directly to perform germline-based case-control association analyses for the discovery of novel cancer predisposition genes.

Combining the cases under scrutiny with external controls for prioritizing disease-predisposition genes is one solution. The best approach to controlling for confounding batch effects caused by the heterogeneity of exome-capture protocols, sequencing platforms, and bioinformatics-processing pipelines would be to re-map all raw sequencing data and then jointly call the genotypes of cases and controls. However, this approach is expensive and requires a lot of storage and a long processing time, which might not be feasible for some research groups. Another solution would be to download jointly called control genotype matrices (if available), merge with case genotype matrices, and apply variant quality control (QC) and filtering to account for batch effects. Alternatively, when the full-genotype data are not available and assuming the pipelines have negligible effects on the association analyses after QC and filtering, one can use publicly available summary counts for case–control association analyses[4]. Here the summary counts refer to the allele or genotype counts of each variant within a cohort. If confounding factors are well-controlled, the summary counts–based strategy is the least expensive and can dramatically increase the sample size of the controls. For instance, gnomAD V2.1 has more than 120,000 WES samples, and gnomAD V3 has more than 70,000 WGS samples[1].

Using high-quality summary counts has led to important supporting evidence in identifying pathogenic germline mutations in adult cancers[5] and pediatric cancers[6]. However, research is lacking on developing a general framework for using high-quality summary counts and evaluating the performance of such a framework.

Several challenges remain when using public summary counts for rare variant association tests. First, because the genotypes of the cases and controls are called separately, the QC and filtering steps might not be consistent if performed separately for cases and controls. To account for such differences, Guo et al.[4] relied on the variant annotation QD (quality score normalized by allele depth); they also used synonymous variants to search for threshold combinations that would remove systematic inflation. One disadvantage of this approach is that it relies on one variant QC metric, instead of more sophisticated variant QC methods, such as VQSR[7].

Differences in the genetic composition of unmatched cases and controls present another challenge. When using publicly available summary counts, researchers often use Fisher's exact test (FET)[4,6] which either treats different ethnicities as a single population or selects samples from one major ethnicity, e.g., samples of European ancestry. Ignoring ethnicity when using public summary counts can result in false positives due to population structure when rare variants have relatively high minor allele frequencies (MAFs) in a specific population. On the other hand, considering a single matched ethnicity might reduce the statistical power. Although a better and more common practice is to use all samples and account for the population structure by adjusting for the principal components (PCs)[8], it is not applicable when only public summary counts are available without individual genotype information.

The inflation factor estimation of the association test when focusing on very rare variants is also important. In genome-wide association studies with common variants, the null distribution of $P$ values is assumed to be continuous and follow a uniform distribution $U(0, 1)$. However, for rare variants, especially those with very low MAFs, assuming a uniform distribution of the $P$ values under the null hypothesis of no association is no longer accurate. For example, in the FET under the null hypothesis of no association, the count of cases with alternate alleles in a $2 \times 2$ contingency table for each gene follows heterogenous hypergeometric distributions, depending on the total number of rare alleles. For the same reason, traditional FDR-control methods such as the BH procedure[9] assuming a uniform distribution of the $P$ values under the null is suboptimal for the discrete count-based test results[10].

Lastly, the most critical challenge when using public summary counts is related to rare variants in high-linkage disequilibrium (LD). In most cases, the independence assumption is probably reasonable because the variants are rare and the chance of observing both variants in one sample is low, unless they are in strong LD. However, when the assumption is violated, either due to high LD or technical artifacts, that can cause false-positive findings or systematic inflation.

In this study, we develop a framework called CoCoRV to prioritize disease-predisposition genes by using public summary counts as controls for rare variant burden tests, making it possible to discover novel genes without sequencing additional healthy control samples. Our framework provides consistent variant QC and filtering, ethnicity-stratified gene-based burden test, accurate inflation factor estimation, powerful FDR control, and an approach to detect high-LD variants using gnomAD summary counts, which is essential for removing false positives. We also evaluate the concordance between using summary counts and full genotypes. In general, the concordance is good, especially for top-ranked genes with a low $P$ value threshold. We focus on genotype calls from WES data throughout this study.

## Results

**Overview of the proposed framework**. To address the challenges in rare variant association tests using public summary counts, we developed a new framework termed Consistent summary Counts based Rare Variant burden test (CoCoRV) (Fig. 1). The input case data can be either full-genotype-based or summary-count-based. The input control data are summary-count data such as from gnomAD. We included a fast and scalable tool to calculate the coverage summary statistics because coverage depth is important in consistent filtering between cases and control[4]. Cases belonging to different ethnicities can be matched to corresponding controls. CoCoRV allows a user to customize criteria for filtering variants based on annotations in the input data. We provided several built-in variant categories based on annotations from ANNOVAR[11] and REVEL scores[12]. For putative pathogenic variants, we included variants annotated with "stopgain", "frameshift_insertion", "frameshift_deletion", and "nonsynonymous" with a REVEL score ≥0.65, which showed empirically good discrimination power for potential pathogenic versus nonpathogenic variants.

CoCoRV ensures that the variant QC and filtering are consistently applied between cases and controls. Only

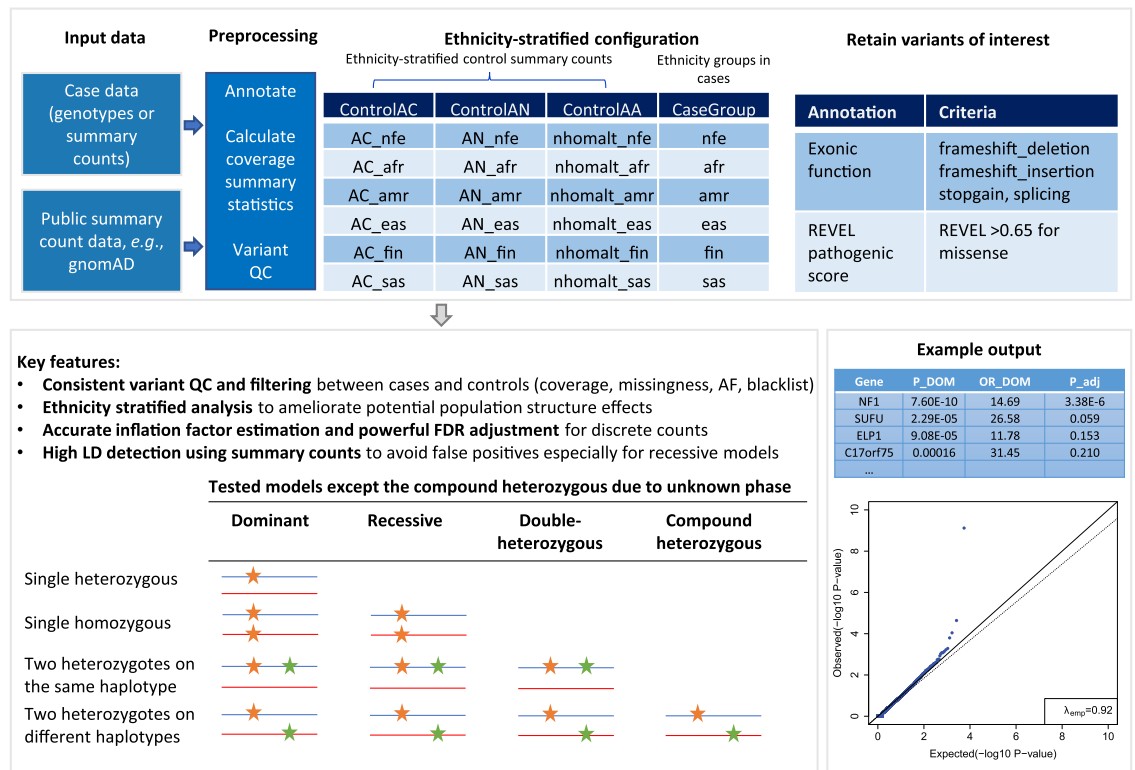

**Fig. 1 Overview of the proposed framework CoCoRV.** The top box describes the main input and preprocessing. The bottom left box describes the key features and the tested models. The bottom right box shows the main output. ControlAC: the annotation for the alternate allele count; ControlAN: the annotation for the total allele count; ControlAA: the annotation for the homozygous alternate genotype count; P_DOM: the raw *P* value of the dominant model using two-sided Fisher's exact test, OR_DOM: the odds ratio of the dominant model; P_adj: the adjusted *P* values for multiple testing.

high-quality variants are retained in the analysis by controlling the coverage depth, missingness of the variants in both case and controls. A blacklist of potentially problematic variants was created based on gnomAD's filtering status and discrepancies in allele frequencies between the WES and WGS platforms (Supplementary Methods). In addition to the FET, CoCoRV includes ethnicity-stratified analysis using the Cochran–Mantel–Haenszel (CMH)-exact test, which mitigates systematic inflations when samples include multiple ethnicities. We computed sample counts in three models (dominant, recessive, and double-heterozygous) within each gene.

Due to the discrete nature of count data, we propose an accurate inflation factor estimation method that is based on sampling the true null distribution of the test statistics (Fig. 2a) and is essential for checking possible systematic inflations. Similarly, the commonly used BH procedure[9] for FDR control assumes a continuous uniform distribution, which is not true for discrete count-based tests. We, therefore, propose to use more powerful resampling-based FDR-control methods or FDR methods directly accounting for the discrete and heterogeneous distribution for the FET and CMH test results (Fig. 2a). The sampled *P* values under the null hypothesis can be used for both inflation factor estimation and resampling-based FDR control.

We introduce a LD-detection method using only gnomAD summary counts to identify high-LD variants (Fig. 2b). It partitions the gnomAD data set into several independent summary-count sets, and then infers the high-LD between variants based on the generated independent summary counts for each ethnicity. Excluding high-LD variants results in a more accurate estimation of the number of samples in each model and reduces false positives when the assumption of independence between variants is violated, especially for the recessive and

double-heterozygous models. Details of the framework and modeling are provided in "Methods". We also provide a side-by-side comparison of the features of our proposed tool and other existing tools for summary-count-based analysis (Supplementary Data 1).

**CoCoRV's inflation factor estimation is unbiased and accurate.** We compared our proposed inflation estimation method with two other methods described by TRAPD[4]. We used the rare variant analysis results of the central nervous system tumors (CNS) and acute lymphoblastic leukemia (ALL) cohorts against our in-house controls under a dominant model to compare these methods (Fig. 3). We also checked their distributions under the simulated null *P* values, where a well-estimated inflation factor should be unbiased and centered around 1. For the CNS cohort tested using FET (Fig. 3a), the inflation factor $\lambda_{emp}$ estimated using empirical null *P* values was 0.96, and those of the other two methods were greater than 1. For the ALL cohort tested using FET (Fig. 3b), all three inflation factors were less than 1. The rightmost plots in Fig. 3 show that $\lambda_{emp}$ was well-calibrated under the null hypothesis, while the other two methods from TRAPD showed either upward or downward biases. The QQ plots based on CoCoRV's method showed almost straight lines along the diagonal, and the other two methods showed clear shifts from the diagonal. In addition to FET, we saw similar patterns using the CMH-exact test (Fig. 3c, d). The estimation of CoCoRV's $\lambda_{emp}$ was stable, even with as few as 100 simulated null *P* values. Therefore, in later estimations of $\lambda_{emp}$ in QQ plots, we used 100 simulated null *P* values per gene, unless otherwise specified. When the null distribution of the inflation factor is of interest, as shown in the rightmost plots, it is better to use at least 1000

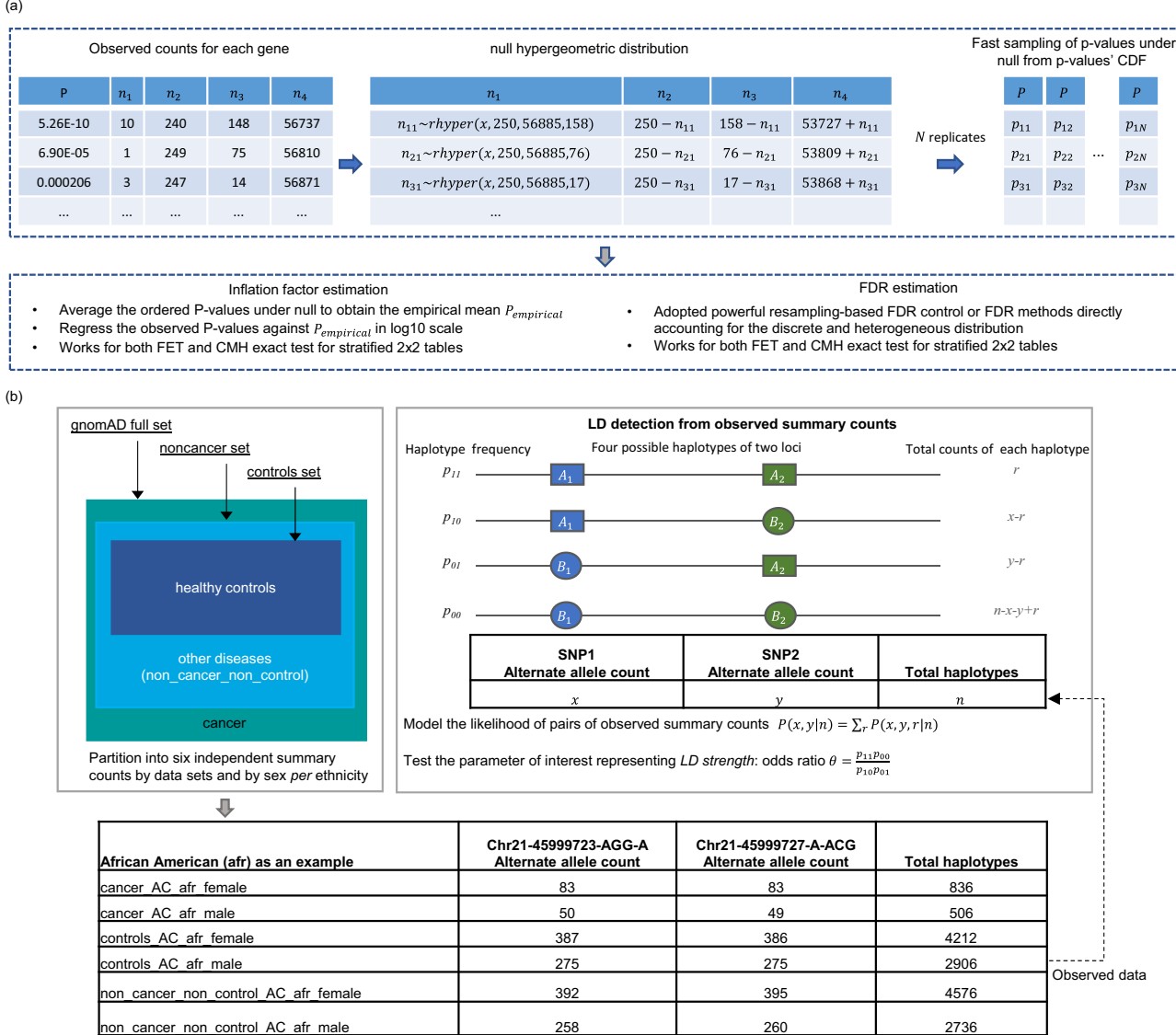

**Fig. 2 The schematic representation of proposed methods in CoCoRV. a** The estimation of inflation factors and FDR. We sample the raw *P* values under the null directly from *P* value's cumulative distribution function (CDF) which is very fast. Either Fisher's exact test or Cochran–Mantel–Haenszel exact test can be used. Either one-sided or two-sided test can be specified. **b** LD detection using gnomAD summary counts. The gnomAD full data sets were partitioned into three independent sets of summary counts and then further partitioned into six independent sets stratified by sex. The six independent summary counts between a pair of variants can be modeled with the LD strength as a parameter, and the LD strength can be tested for LD detection. The table at the bottom shows an example of the six independent summary counts between two variants of high LD in the African American ethnicity group.

replications for a more stable result. Because of the better performance of the proposed estimation, we applied it to nearly all subsequent analysis results in this study. The run time of our inflation factor estimation is less than a minute for sampling 1000 replicates of ~20,000 genes, considerably fast because of the direct computing of the cumulative distribution function (CDF) of *P* values under the null hypothesis.

**Concordance of rare variant association between using summary counts and full genotypes.** We compared the concordance between the result obtained using summary counts from separately called controls and that using jointly called full genotypes. For this analysis, we generated separately called control summary counts using an in-house control cohort of 8175 WES and treated the jointly called case–control full genotypes as the ground truth (Supplementary Fig. S1). To our knowledge, this is the first direct

comparison between analysis results using summary counts of separately called cases and controls and that using jointly called full genotypes. For this comparison, we focused on the commonly used dominant model. We used an AF threshold of 1*e*−3 for the CNS and ALL cohorts and for all comparisons.

We first compared the sample counts with defined pathogenic variants for each gene. For cases, we compared and tabulated the sample counts with defined potential pathogenic variants from using jointly called full genotypes and that from using summary counts into a cross-classification/contingency table (Fig. 4a, b). For both the CNS and ALL cohorts, the concordance between using full genotypes and CoCoRV was very high (Pearson's correlation *r* > 0.98). Most of the discordances were within one count. For controls, because the number of samples with defined potential pathogenic variants varied in a large range, we used the scatter plot of those numbers to illustrate the concordance between using full genotypes and using summary

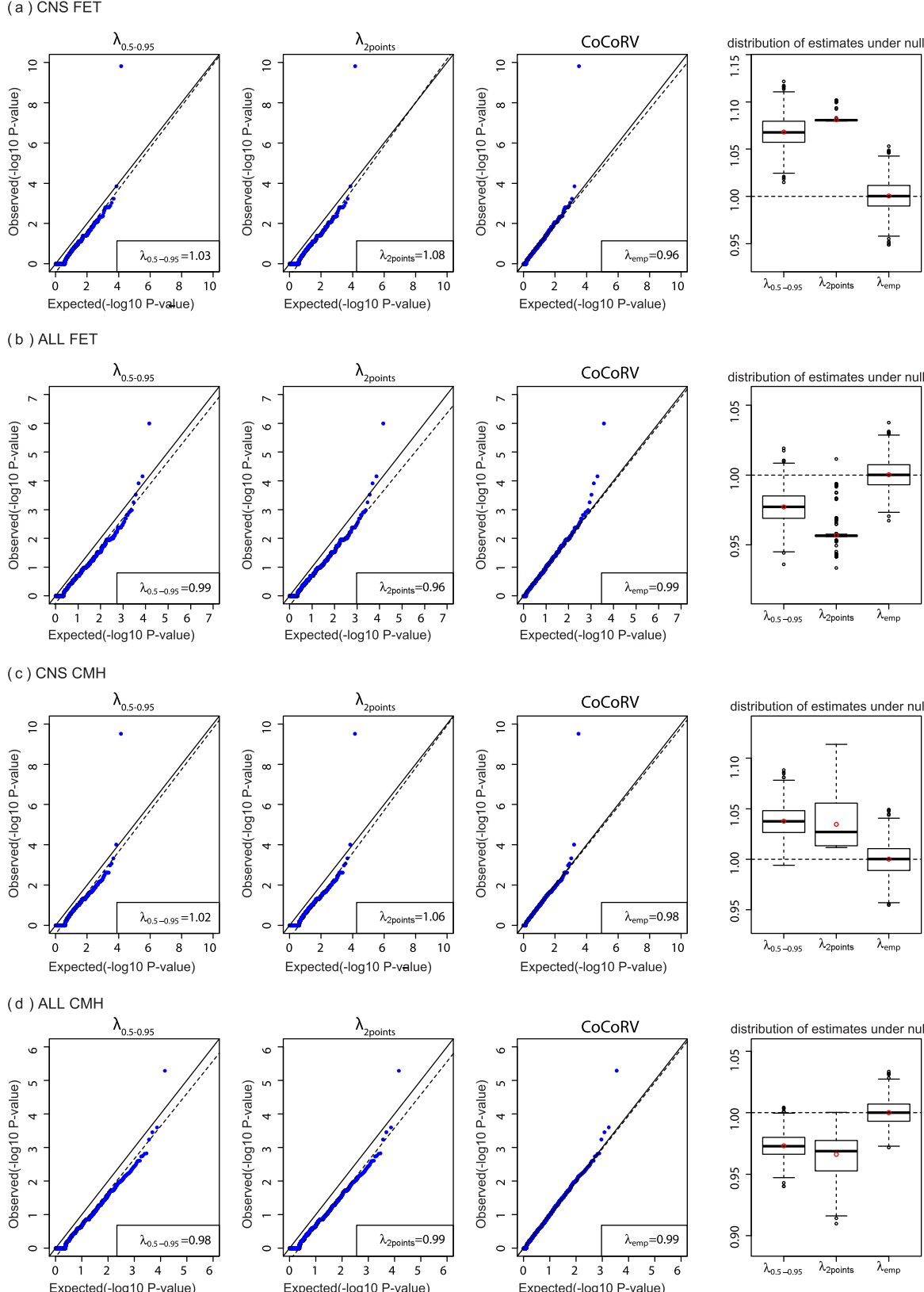

counts (Fig. 4c, d). The correlation between using full genotypes and CoCoRV was also very high ($r = 0.996$). The high correlation of the raw counts shows that the proposed summary counts-based framework maintained a high concordance with the jointly called full-genotype-based framework. We also examined the top genes of the association tests between using jointly called full genotypes and separately called summary counts (Fig. 4e, f). In general, the percentage of top overlapped genes was about 70% or higher, especially for the more stringent $P$ value thresholds. We also performed ethnicity-stratified analyses for both frameworks and found similar overlapping patterns (Supplementary Fig. S2). Finally, the QQ plots using jointly called full genotypes and

**Fig. 3 Comparisons of different inflation factor estimation methods.** The first three columns show the inflation factors estimated using different methods. The dashed lines are the fitted lines in the inflation factor estimation. The fourth column shows boxplots of the estimated inflation factors using simulated $P$ values under the null hypothesis ($n = 1000$ independent replicates). The red circle indicates the mean value, the center line indicates the median value, the bounds of the box covers data between 25th and 75th percentile. The outliers are points above or below 1.5 times the interquartile range starting from the 75th or 25th percentile. If there is no outlier, the whisker defines the minima or maxima, otherwise it is 25th percentile minus 1.5 times interquartile range or 75th percentile plus 1.5 times the interquartile range. **a** Results based on the central nervous system (CNS) cancer cohort and the two-sided Fisher's exact test (FET). **b** Results based on the acute lymphoblastic leukemia (ALL) cohort and the two-sided FET. **c** Results based on the CNS cohort and the two-sided Cochran–Mantel–Haenszel exact test (CMH) stratified by ethnicities. **d** Results based on the ALL cohort and the two-sided CMH stratified by ethnicities.

separately called summary counts were also similar, with correlations between $-\log_{10} P$ values greater than 0.9 (Fig. 4g, h).

We also investigated the coverage depth cutoff in QC. The coverage depth cutoff of 10 shows the best concordance measured by correlations of qualified sample counts in cases or controls, and correlations of the association test $P$ values (Supplementary Data 2). The best concordance of using the depth cutoff 10 could be related to the QC of genotypes: we keep high-quality genotypes with $DP \geq 10$ (DP is the number of informative reads for each sample in the VCF file), which is consistent with the QC used in gnomAD for summary allele counts. As the coverage cutoff increases, the variant quality will increase too, but we might risk missing good quality variants if the coverage cutoff is too high, especially for some regions not well covered due to high GC contents or other sequencing features. We also calculated the size of qualified regions with different coverage cutoffs on gnomAD's whole-exome coverage data (Supplementary Fig. S3). The coverage cutoff of 10 seems to be a good trade-off between variant quality and the size of qualified regions being retained.

**FDR methods accounting for discrete counts show good FDR control and improved power.** Two adopted resampling-based methods: RBH_P and RBH_UL, and one recently developed method designed for discrete distributions ADBH.sd[10] show good FDR control in general and similar improved power for FET-based test results, even though RBH_P has slightly inflated FDR in some settings (Supplementary Fig. S4). Compared to BH, the relative power increases of RBH_UL range from 11 to 27% (Supplementary Data 3). Filtering the genes with few rare allele counts (BH_T2 and BH_T3) shows little power increase. When applied to the CNS association test results using gnomAD as the control (threshold <0.2), the three methods accounting for the discrete counts all identify the established genes *NF1*, *SUFU*, *ELP1* as significant, while the BH method only identifies *NF1* as significant (Supplementary Data 4). For resampling-based FDR control, the time cost is within a minute, similar as that in inflation estimation because sampling the $P$ values under the null hypothesis is the main time-consuming part. For ADBH.sd, the time cost is even smaller because only $P$ values' CDF is needed without sampling $P$ values under the null.

**The proposed LD-detection method has high power in detecting high-LD rare variants and accurately identifies MNVs in gnomAD.** We first evaluated the type I error and statistical power of the proposed LD test by using simulated data consisting of six independent sets of summary counts (Supplementary Fig. S5a). The type I error was well-controlled. The power of detecting LD increased as the LD strength and the AF increased. The LD test was well powered for detecting strong positive LDs but not for detecting weak LDs. When full genotypes are available, the proposed test can also be used because the full-genotype-based LD test is a special case of our proposed method, where the total haplotypes is 2. We evaluated our method in

simulations under the full-genotype setting and compared it with the *ld* function in snpStats[13] which is designed for LD detection using full genotypes. CoCoRV and snpStats have almost the same results (Supplementary Fig. S5b) and the type I error was well-controlled. The power increased remarkably compared to using summary counts, which was expected given that more information is available when using full genotypes.

As one application and a validation of the proposed LD-detection method, CoCoRV was applied to detect high-LD multinucleotide variants (MNVs)[14] without access to the sequencing reads information. A MNV is defined as a cluster of two or more nearby variants on the same haplotype in an individual. Because variants within a MNV are close to each other, they are very likely transmitted together thus exhibiting high-LD in a population. We applied our proposed LD test to the whole exomes of gnomAD data to detect rare variant pairs of high LD and compare them with the reported gnomAD MNVs detected using sequence reads. Although high-LD variants and MNVs are not the same, they have similarities when restricted to rare variants with distances ≤2 BPs (Table 1). For each ethnicity and each variant, we extracted the gnomAD exome count data and generated six independent sets of summary counts using the cohort and sex information. Then we applied the LD test on pairs of variants annotated with certain functions, excluding synonymous variants.

The detection results in gnomAD *per* ethnicity group are summarized in Table 1. In total, ~10 million tests of variant pairs were performed. By controlling the FDR at 0.05, we detected nearly 10,000 coding variant pairs that are in LD (Supplementary Data 5). Most of the detected variant pairs with FDR < 0.05 were high-LD variants (odds ratio >150). We also checked the overlap between high-LD variants and reported gnomAD MNVs. Because the reported gnomAD coding MNVs are pairs of SNVs within 2 BPs, we restricted the detected pairs to high-LD (odds ratio >150) SNVs for both variants and the distance within 2 BPs. About 90% of them are reported in the gnomAD MNV dataset (either from exomes or genomes) using read-based MNV detection[14]. We then manually checked the read haplotype information from the gnomAD website (https://gnomad.broadinstitute.org/) for those 116 pairs not reported as gnomAD MNVs. There were 85 unique pairs if not distinguishing ethnicities, and 65 (76.4%) had read information. Of the 65 pairs, 60 (92.3%) had supported reads showing that they are on the same haplotype. The reason that these pairs were not reported as gnomAD MNVs might be due to other filtering criteria used. Reported gnomAD MNVs had alternate alleles verified on the same haplotype, which provides strong evidence that the pairs detected through our LD test are true high-LD variants because the likelihood of two independent rare alleles appearing on the same haplotype is very low. Although there was a peak when the distance was <10 BPs, there was also a large mass when it was >10 BPs (Supplementary Fig. S6), emphasizing the importance of detecting high LD with relatively long distance. In general, the recombination rate decreased as the physical distance increased, indicating that even though the physical distance was large, the recombination

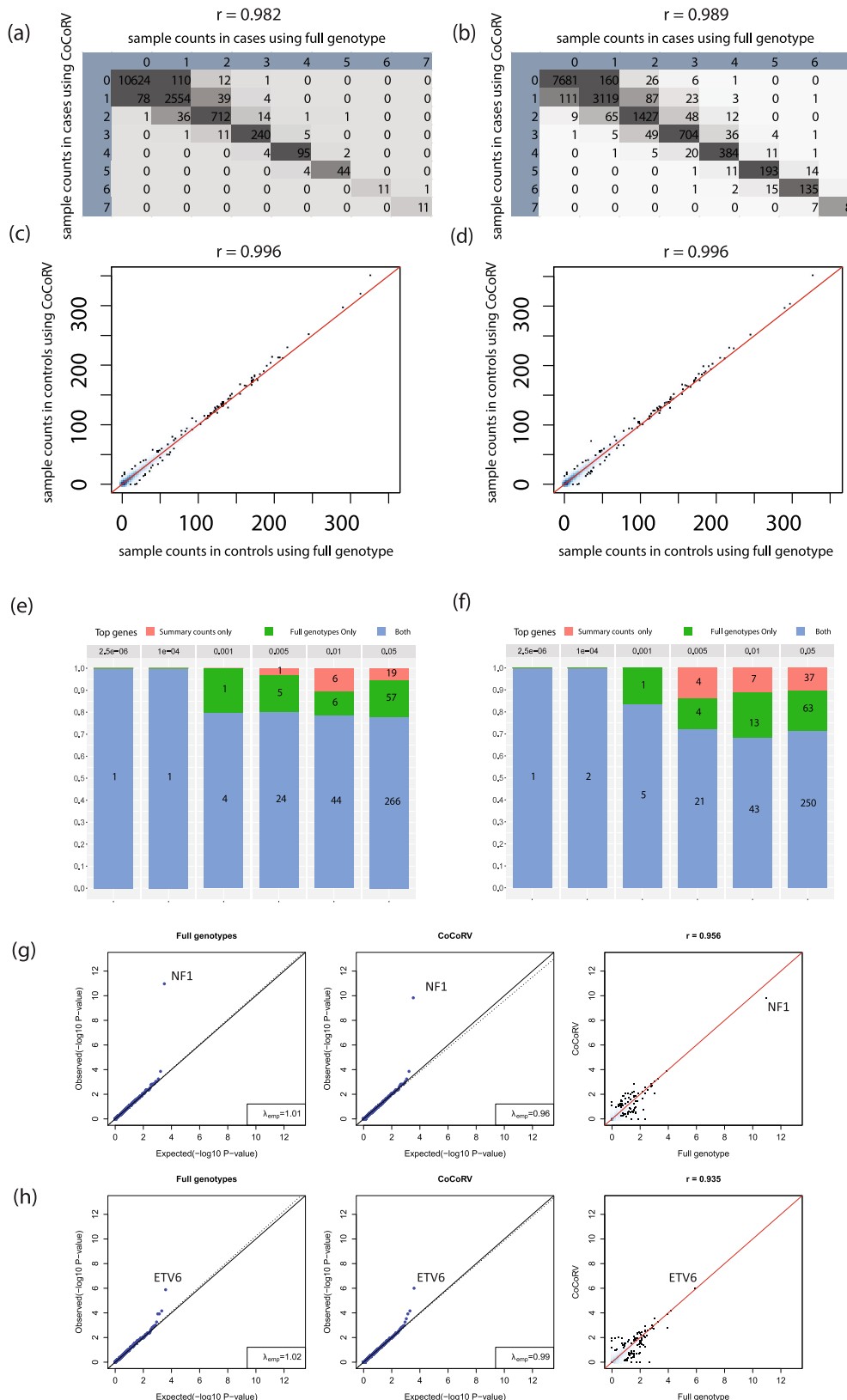

frequency between the two variants can still be low. Over relatively long distances (e.g., >150 BPs), high-LD variants could not be detected by examining the sequencing reads data (usually ~150 BPs) but could be detected using CoCoRV.

The above analyses used the stringent threshold FDR < 0.05 to achieve high precision but could sacrifice the recall rate. We also

used different *P* value thresholds and evaluated the recall and precision by treating the reported gnomAD MNVs (≤2 BPs) as the ground truth, though some high-LD MNVs detected were not reported in the gnomAD MNVs, as described above. In total, there were 1780 reported gnomAD MNVs among our tested variant pairs (Supplementary Data 6). If we chose 0.01 as the

**Fig. 4 Comparisons of concordance between using separately called summary counts and jointly called case–control full genotypes. a, b** Cross-classification tables between using jointly called full genotypes and using separately called summary counts across all genes on the number of samples carrying defined potential pathogenic rare alleles in the CNS (**a**) and ALL (**b**) cohorts. The column ID represents the number of samples calculated using full genotypes, and the row ID represents the number of samples calculated using summary counts. Only counts ≤7 are displayed. Genes with zero samples carrying rare alleles among all cases and controls, calculated by using the full genotypes and summary counts, are excluded. **c, d** Scatter density plots of the number of control samples from the CNS (**c**) and ALL (**d**) cohorts carrying qualified rare alleles, as determined by using full genotypes and summary counts. **e, f** Comparisons of top genes in the CNS (**e**) and ALL (**f**) cohorts between using jointly called full genotypes and separately called summary counts. The bar heights show the percentage, and the numbers within each bar show the absolute number of genes. **g, h** QQ plots of rare variant association results using jointly called full genotypes (left panels), separately called summary counts (middle panels), and scatter plots of –log₁₀ P values between using jointly called full genotypes and separately called summary counts. Analyses results are from the CNS (**g**) and the ALL (**h**) cohorts. All P values are raw P values from the two-sided Fisher's exact test.

**Table 1 LD detection of annotated functional variants within each gene by using gnomAD summary counts.**

| Ethnicity | Sample size | Pairs tested | FDR < 0.05 | High LD[a] | MNV (≤2 BPs) | Reported in gnomAD MNV[b] |
|---|---|---|---|---|---|---|
| nfe | 56,885 | 4,760,061 | 1962 | 1962 | 311 | 278 (0.89) |
| afr | 8128 | 1,166,251 | 3766 | 3762 | 291 | 271 (0.93) |
| amr | 17,296 | 2,171,082 | 1948 | 1948 | 200 | 181 (0.91) |
| eas | 9197 | 483,555 | 923 | 923 | 136 | 118 (0.87) |
| sas | 15,308 | 1,407,370 | 971 | 971 | 119 | 97 (0.82) |
| fin | 10,824 | 177,390 | 511 | 511 | 79 | 75 (0.95) |

*nfe* non-Finnish European, *afr* African American, *amr* Admixed American, *eas* East Asian, *sas* South Asian, *fin* Finnish, *BPs* base pairs, *FDR* false discovery rate, *MNV* multinucleotide variant.
[a]High LD was defined as estimated odds ratio >150.
[b]The union of gnomAD coding MNV and genome MNV with distance ≤2 BPs. The numbers within the parentheses are the proportions of our detected MNV (≤2 BPs) reported in the gnomAD MNV dataset.

P value threshold, 1635 of 1976 detected were true, corresponding to precision 0.83 and recall 0.92.

To further demonstrate the accuracy of our proposed LD-detection method, we compared the LD-detection results using either pooled summary counts or individual genotypes from our constructed in-house 8175 synthetic controls. Specifically, in each ethnicity, we randomly split the control dataset into three datasets with sample size ratio 1:2:3, then we stratified each of the three dataset by sex, generating six independent datasets. For each of the six dataset, we pooled the alternate allele counts and the total allele counts, similar as that from gnomAD, and tested LD between variant pairs using CoCoRV. For individual genotypes, we used the *ld* function in the R package snpStats[13] to test each LD pairs within each ethnicity. We focused on variants with gnomAD maximal allele frequency among ethnicities <0.1 and the selection criteria of variants based on annotations were the same as that in the LD scan for gnomAD. We consider 45,626 LD pairs detected using individual genotypes (FDR < 0.05) as the ground truth. In total, 6694 LD pairs were detected (FDR < 0.05) using summary counts, where 8 were false positives, indicating high precision (99.8%) of CoCoRV. The lower power using summary counts (14.65%) is consistent with our power simulations, suggesting the power advantage of using full genotypes when they are available. For the true positives detected by CoCoRV, the estimated LD measure $r^2$ between using summary counts and using full genotypes were highly correlated (correlation = 0.853) (Supplementary Fig. S7a). After stratifying detected LD pairs by alternate allele frequencies within the controls and the estimated LD measure $r^2$ from using full genotypes (Supplementary Fig. S7b), we observed that summary counts-based LD detection mainly detect those high-LD pairs with decent alternate allele counts (e.g., >10).

Next we validated the detected LD pairs from gnomAD using the 8175 controls with individual-level genotypes. Out of 10,081 LD pairs from gnomAD (FDR < 0.05 within each ethnicity), 7290 (72.3%) variant pairs can be tested with at least one alternate allele from at least one variant of the pair in the individual-level controls. We used a P value threshold $5 \times 10^{-4}$, which

corresponds to 3.6 expected false positives out of 7290 tests under the null hypothesis of no LD. Of the 7290 variant pairs tested, 6989 (95.9%) passed the P value threshold $5 \times 10^{-4}$ (Supplementary Data 5). All pairs that passed the threshold except two have odds ratio larger than 150. These results show that our proposed method of detecting LD from gnomAD using summary counts is accurate or has high precision.

We also compared the LD detected using gnomAD and LD detected using available individual data (data from the 1000 Genomes Project and the larger 8175 constructed control data). For individual genotype-based data, we used the *ld* function from the R package snpStats. We focused on variants with gnomAD maximal allele frequency among ethnicities <0.1. LD test were performed within each ethnicity and the selection criteria of variants based on annotations were the same as that in the LD scan for gnomAD. We used FDR threshold 0.05 for each ethnicity and focused on high LD (odds ratio >10). We stratified the significant LD pairs based on their alternate allele frequencies and calculated the contributions of each data source (Supplementary Fig. S8). For relatively frequent variants, e.g., the AF range [0.01, 0.1), the unique contribution of gnomAD-based LD detection is small (<3%), however, as the AF range becomes lower, the contribution of gnomAD-based LD detection becomes substantial. For example, within the AF range [5e−4, 0.001), over 30% of the significant LD pairs can only be discovered using gnomAD-based approach if compared with that using the 1000 Genomes Project's individual-level data, the unique contribution is still about 25% if compared with that using the larger 8175 constructed individual control data. These substantial contributions are likely due to the large sample size of different ethnicities in gnomAD. The unique contribution from gnomAD can be even higher when considering each ethnicity individually (Supplementary Fig. S9). For example, in the AF range [5e−4, 0.001), the unique contribution of gnomAD-based approach can be over 35% for the East Asian and South Asian, and about 80% for the Finnish population, even when compared with that using the larger 8175 individual-level control data. On the other hand, the unique contribution from that using full genotypes is usually

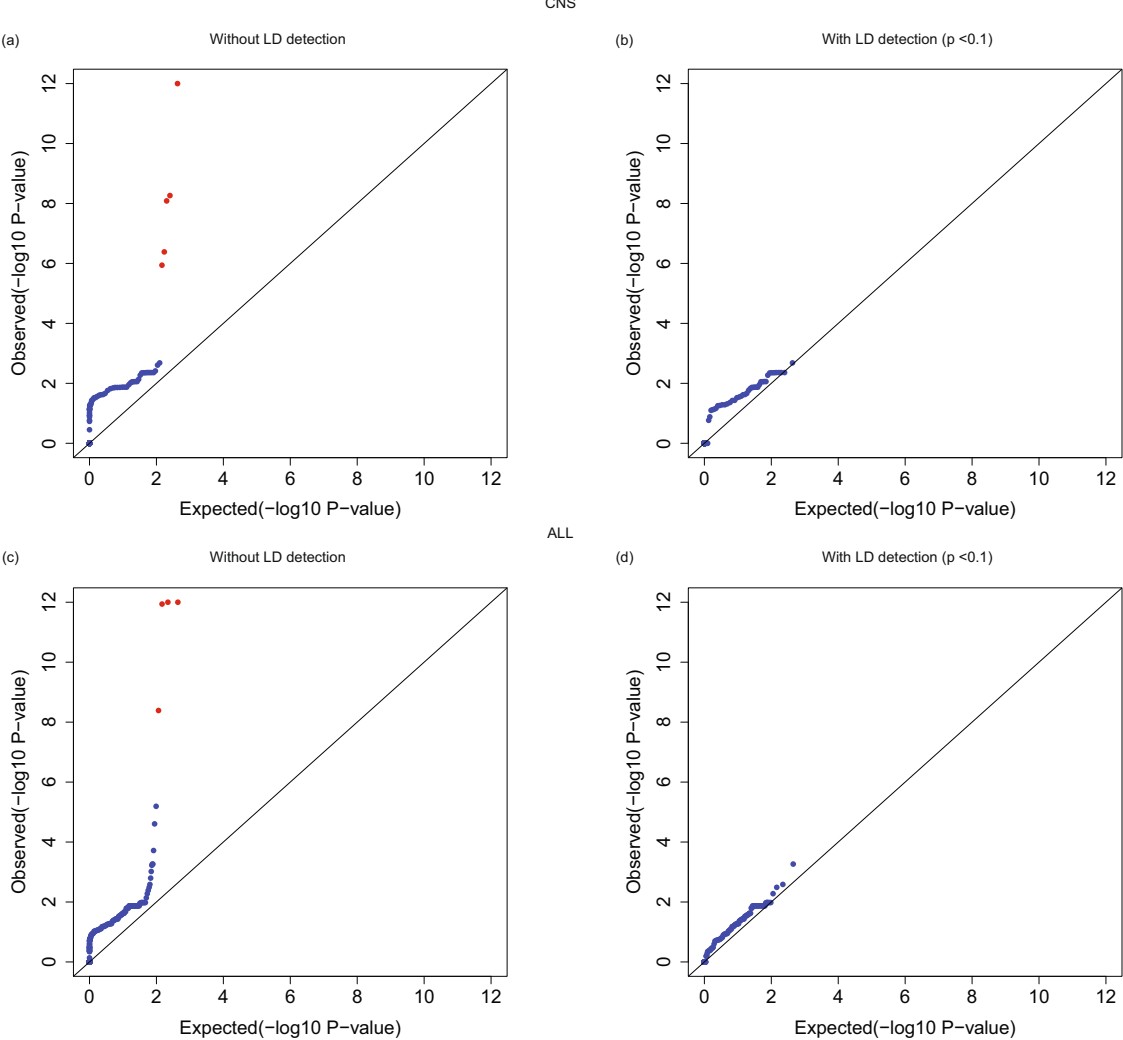

**Fig. 5 Employing LD detection removes false positives under the recessive model. a, c** QQ plots of association tests under the recessive model without employing LD detection in the CNS and ALL cohorts, respectively. **b, d** QQ plots of association tests under the recessive model after employing LD detection with a *P* value threshold of 0.1. Red dots in (**a**) and (**c**) are false-positive genes that pass the exome-wide significance threshold (*P* < 2.5e-6). The *P* values are raw *P* values from the two-sided Cochran–Mantel–Haenszel (CMH) exact test.

larger, which is also consistent with our power simulation results. Considering the two different data sources can complement each other, we merged the LD-detection results from both data sources and use them for checking LD under the recessive models.

**Detecting high-LD variants improves count estimation and removes false positives in the recessive and double-heterozygous models.** We performed a simulation of count estimation by using summary counts from a set of variants, including independent and correlated variants (Supplementary Fig. S10). Our results showed that the count estimation results from using CoCoRV were more accurate than those from using TRAPD. TRAPD overestimated the counts for all three models when high-LD variants were present, most likely because TRAPD was designed to be conservative for a one-sided FET to mitigate the violation of the assumption of independent variants. This overestimation could result in an inflated type I error, and the potential loss of power if a two-sided test is used. CoCoRV corrected for high-LD variants and resulted in a much better estimation.

We also compared the association results under the recessive model, with or without the LD test with an AF threshold 0.01 and using gnomAD summary counts as controls for both the CNS and

ALL cohorts (Fig. 5). When there was no LD check for cases with two heterozygous variants, we found several false positives with very low *P* values (Fig. 5a, c). When variants in LD were excluded from the count of case samples (Fig. 5b, d), those false positives were removed (Supplementary Data 7). One example is the variant pair (3-37089130-A-G, 3-37089131-A–C) in gene MLH1, which is also identified as an MNV from gnomAD. Applying the LD test also removed false positives with very low *P* values under the double-heterozygous model (Supplementary Fig. S11). We noticed that the QQ plots under the recessive model were not as well-calibrated as those under the dominant model, even after applying the LD test. The reason for this result could be that not all double-heterozygous variants with positive LD were detected. Another factor to consider is that the recessive and double-heterozygous models are sensitive to differences in LD structures between cases and controls. Stratifying samples into different ethnicities might still retain some LD differences due to the population substructure within each of the six ethnicities. For example, African/afr and Latino/amr are two main admixed populations, and different proportions of ancestries can change the LD structure. Although there is no perfect solution, our CoCoRV method represents an advance in using only summary counts as controls for rare variant association tests.

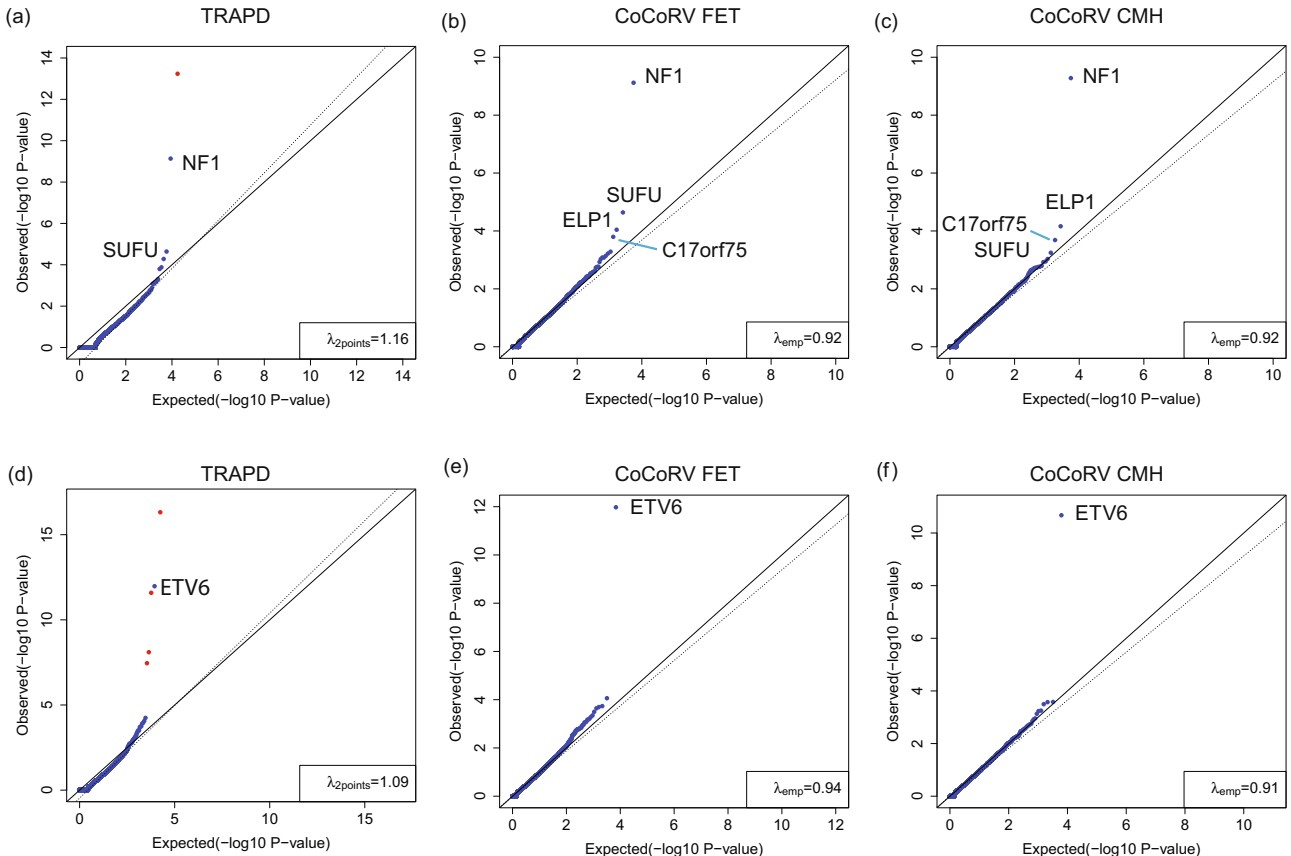

**Fig. 6 CoCoRV and TRAPD analyses of the CNS and ALL cohorts.** We used gnomAD summary counts as controls. **a–c** The analyses of the CNS cohort using TRAPD (**a**), CoCoRV with the two-sided Fisher's exact test (FET) (**b**), and CoCoRV with the two-sided Cochran–Mantel–Haenszel (CMH) exact test (**c**). **d–f** The analysis of the ALL cohort using TRAPD with the two-sided FET (**d**), CoCoRV with the two-sided FET (**e**), and CoCoRV with the two-sided CMH-exact test (**f**). Red dots in (**a**) and (**d**) indicate false-positive genes that pass the exome-wide significance threshold ($P < 2.5e\text{-}6$). All $P$ values are raw $P$ values without multiple testing adjustments.

**CoCoRV analysis of the CNS and ALL cohorts**. We applied CoCoRV to two St. Jude pediatric cancer cohorts (CNS and ALL) and used gnomAD summary counts as controls under a dominant model. For a comparison, we also applied TRAPD to the two cohorts. We used the all-pooled $AC$ and $AN$ to generate the counts per gene because TRAPD is hard-coded to use the annotation $AC$ and $AN$ to generate case–control statistics. CoCoRV was flexible to specify any annotated fields, such as non-cancer subsets, or any population-specific counts from gnomAD. Both SNVs and short indels were considered. For TRAPD, using the option "--pass" shows less false positives than using the QD-based filtering (Supplementary Data 8), therefore we used the former for comparison with CoCoRV. For the CNS cohort, TRAPD reported an inflated $\lambda_{2points}$ (Fig. 6a). Two genes passed the exome-wide significance ($P < 2.5 \times 10^{-6}$) using TRAPD. One was a false positive and the other was the known glioma causal gene *NF1*. For CoCoRV, it showed no inflation in either the pooled counts from all ethnicities (Fig. 6b) or the stratified analysis using CMH (Fig. 6c). One gene passed the exome-wide significance, which was the known causal gene *NF1*. After manual variant checking, the false positive from TRAPD was caused by inconsistent filtering: the variant has FILTER status PASS in cases and therefore included in cases but has the failed status (failed the random forest-based QC) in gnomAD therefore not included in controls (Supplementary Data 8). For the ALL cohort, five genes passed exome-wide significance using TRAPD and four of them were false positives. The inflation factor $\lambda_{2points}$ was inflated (Fig. 6d). In contrast, CoCoRV showed no obvious systematic

inflation, and identified the known causal gene *ETV6* as the only exome-wide significant gene. The CMH-based analysis had better calibration at the tail than did the FET-based analysis (Fig. 6e, f). The cause of the false positives from TRAPD was either inconsistent filtering between cases and controls or low-quality variants which showed large differences in AF between gnomAD WES and WGS data (Supplementary Data 8).

*NF1* was detected in the pediatric CNS cohort and had relatively similar $P$ values across different scenarios, no matter which summary counts-based controls were used. For the ALL cohort, the much larger sample size of gnomAD dramatically increased the significance of *ETV6*, from $10^{-6}$ to $10^{-12}$ (Table 2). The relatively stable $P$ values for *NF1* in the CNS cohort were most likely due to the adequately large counts in the contingency table using the in-house–constructed summary counts. The benefit of increasing the number of controls also diminished after a certain level, while the number of cases was fixed (Table 2). In contrast, *ETV6* had 0 or 1 qualified control sample with rare alleles in the in-house controls, had more significant $P$ values by using gnomAD as controls, benefited from the improved quantification of the AFs when the control sample size increased substantially.

For the CNS cohort, the top four genes were *NF1*, *ELP1*, *C17orf75*, and *SUFU* (Fig. 6). *NF1* and *SUFU* are well-established causative genes for pediatric brain tumors[15]. *ELP1* was recently identified as a predisposition gene of medulloblastoma, one type of pediatric brain tumors[6]. This makes *C17orf75* an interesting candidate. From the GTEx portal, two of the top three tissues

**Table 2 Statistics of the top genes by using different methods and alternate allele frequency thresholds.**

| Cohorts and genes | Method[a] | $n_1$ | $n_2$ | $n_3$ | $n_4$ | P value |
|---|---|---|---|---|---|---|
| AF ≤8e-5 | | | | | | |
| CNS ($n = 336$) NF1 | Jointly called full genotype | 12 | 324 | 6 | 8169 | 1.77e-13 |
| | CoCoRV using in-house controls | 11 | 325 | 6 | 8169 | 3.09e-12 |
| | CoCoRV using gnomAD as controls | 11 | 325 | 262 | 125,486 | 2.82e-10 |
| ALL ($n = 958$) ETV6 | Jointly called full genotypes | 6 | 952 | 0 | 8175 | 1.31e-06 |
| | CoCoRV using in-house controls | 7 | 951 | 1 | 8174 | 9.96e-07 |
| | CoCoRV using gnomAD as controls | 7 | 951 | 5 | 125,743 | 1.06e-12 |
| AF ≤1e-3 | | | | | | |
| CNS ($n = 336$) NF1 | Jointly called full genotype | 12 | 324 | 11 | 8164 | 1.08e-11 |
| | CoCoRV using in-house controls | 11 | 325 | 11 | 8164 | 1.48e-10 |
| | CoCoRV using gnomAD as controls | 11 | 325 | 289 | 125,459 | 7.60e-10 |
| ALL ($n = 958$) ETV6 | Jointly called full genotypes | 6 | 952 | 0 | 8175 | 1.31e-06 |
| | CoCoRV using in-house controls | 7 | 951 | 1 | 8174 | 9.96e-07 |
| | CoCoRV using gnomAD as controls | 7 | 951 | 5 | 125,743 | 1.06e-12 |

AF alternate allele frequency, ALL acute lymphoblastic leukemia, CNS central nervous system.
[a]The CoCoRV results were calculated based on the FET.
$n_1$: number of cases with potential pathogenic alleles; $n_2$: number of cases without potential pathogenic alleles; $n_3$: number of controls with potential pathogenic alleles; $n_4$: number of controls without potential pathogenic alleles.

with highest median expression of *C17orf75* are the cerebral hemispheres and the cerebellum (Supplementary Fig. S12). Further validation is needed to confirm the association of *C17orf75* with pediatric brain tumors.

ProxECAT[16] is another recently proposed method for performing summary-count-based analysis. It pools the counts of functional and non-functional alleles from cases and controls to form a contingency table, which is used for an association test. We implemented this method within CoCoRV and applied it to the CNS and ALL cohort. ProxECAT showed slight inflations (inflation factor 1.05 for ALL, and 1.08 for LGG). The known gene *NF1* and *ETV6* reached the exome-wide significance level and there were no false positives reaching the exome-wide significance (Supplementary Data 8). For the known causal genes *NF1*, CoCoRV shows more significant *P* values ($P = 7.6 \times 10^{-10}$) than ProxECAT ($P = 9.2 \times 10^{-8}$), the trend is the same for *ETV6* too (Supplementary Data 9).

We also explored the recessive models in the association test using both TRAPD and CoCoRV (Supplementary Data 10). When the LD was not accounted for, for both TRAPD and CoCoRV, all genes that passed the exome-wide significance were due to the high LD between variants, therefore considered as false positives. However, when the LD was accounted for in CoCoRV, there was no gene passing the exome-wide significance. This suggests that for the CNS and ALL, the association signal is mainly driven by the dominant model, although it is also possible that we do not have enough power for the recessive models with the current sample size of cases.

**Analysis of germline samples from The Cancer Genome Atlas (TCGA).** We applied CoCoRV to the TCGA WES datasets of glioblastoma multiforme (GBM) and low-grade glioma (LGG). Brain tumors can be classified into grades I–IV based on standards set by the World Health Organization. The TCGA LGG data consist of grades II and III, and the GBM data are considered grade IV. The processing of raw reads alignment, variant QC, ethnicity, and relatedness inference were the same as those used for the CNS and ALL cohorts. After removing related samples, 325 GBM samples and 483 LGG samples remained. We used CoCoRV to perform summary-count-based association tests separately for GBM and LGG samples. We selected the non-cancer summary counts from gnomAD as controls and used the CMH ethnicity-stratified test. The threshold of AF and the AF_popmax (the maximal AF among ethnicities from the

**Table 3 The top candidate genes from the TCGA GBM cohort and statistical evidence from the TCGA LGG cohort and the PCGP HGG cohort.**

| | TCGA GBM | | TCGA LGG | PCGP HGG |
|---|---|---|---|---|
| chr | Genes | P value | P value | P value |
| 1 | TTC39A | **7.30e-06** | 1 | – |
| 17 | TP53 | **5.60e-05** | **0.02** | **0.01** |
| 12 | ELK3 | **3.80e-04** | 1 | – |
| 7 | ABCB8 | **4.62e-04** | **0.01** | **0.005** |
| 1 | CELA2B | **1.13e-03** | 0.14 | – |

chr chromosome, FDR false discovery rate, GBM glioblastoma multiforme, HGG high-grade glioma, LGG low-grade glioma, PCGP Pediatric Cancer Genome Project, TCGA The Cancer Genome Atlas. P values less than 0.05 are in bold.

gnomAD annotation) were set to $1e-3$. The same criteria were used to define potential pathogenic variants, as in the pediatric CNS and ALL cohorts. We observed no inflation in the QQ plot (Supplementary Fig. S13). Table 3 shows the top five genes from the results of the GBM cohort. We further tested these five genes in the TCGA LGG cohort and found that two genes, *TP53* and *ABCB8*, were significant ($P < 0.05$) (Table 3). As an independent statistical validation, we used 85 pediatric high-grade glioma WES samples from the PCGP project[17] as cases and independent 8525 controls, including additional non-cancer in-house samples and samples from the ADSP and the 1000 Genomes Project. We used the full-genotype-based analysis in this validation for the best control of confounding effects. The top 20 PCs were included as covariates, and the VT method implemented in EPACTS[18] was used for the rare variant association test. The *P* values of *TP53* and *ABCB8* were significant ($P = 0.01$ and 0.005, respectively) (Table 3). *TP53* is a known tumor-suppressor gene and a predisposition gene for many cancers, including adult and pediatric brain tumors[15,19]. Huang et al.[5] performed a germline-based association analysis of TCGA data focusing on 152 candidate genes including *TP53*; however, *TP53* was not listed as significant for GBM. This false negative is most likely due to their analysis design: they used one cancer versus all other cancers in TCGA, and *TP53* is enriched in multiple cancers so the enrichment in GBM is not detected. *ABCB8* is an interesting candidate gene associated with brain tumors that encodes an ATP-binding subunit of the mitochondrial potassium channel. Potassium channels

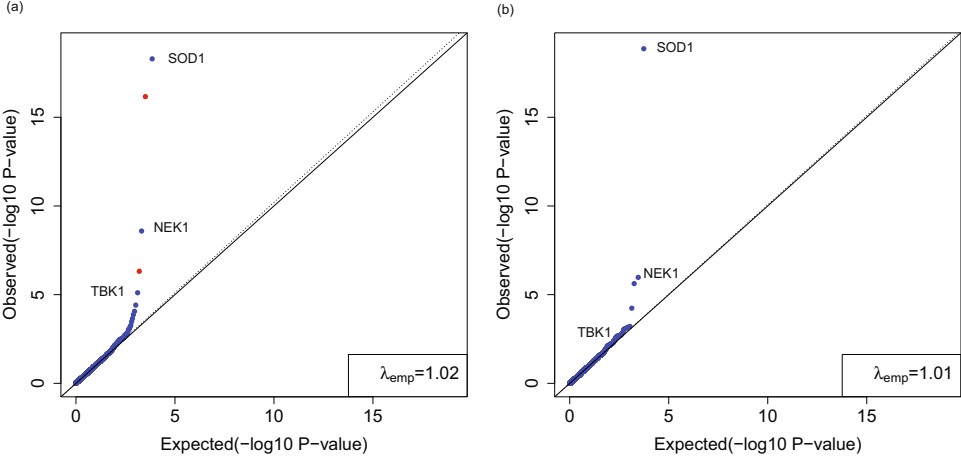

**Fig. 7 Rare variant analysis of the Caucasian samples from the ALSDB data using summary counts for both cases and controls. a** CoCoRV analysis with the two-sided Fisher's exact test using control summary counts from the gnomAD non-Finnish Europeans. The red dots are false positives after a manual check on the variants contributing to the test statistics: the first false positive is due to a large cluster of indels in the study-specific case and control samples; the second is due to a variant with strong Hardy–Weinberg disequilibrium in both study-specific cases and controls. **b** CoCoRV analysis with the two-sided Fisher's exact test using control summary counts from the study-specific 8186 Caucasian control samples. All *P* values are raw *P* values without multiple testing adjustment.

can promote cell invasion and brain tumor metastasis[20]. They also have an important role in brain tumor biology[21]. Per the GTEx portal, the cerebellum has the highest median expression of *ABCB8* (Supplementary Fig. S12). Further experiments are crucial to confirm whether deleterious *ABCB8* mutations increase the risk to adult brain tumors.

**Analysis of an amyotrophic lateral sclerosis study using case only summary counts**. We used publicly available case summary counts and gnomAD summary counts for rare variant analysis. Specifically, we downloaded the summary counts of an ALS study[22] from the ALSDB website (http://alsdb.org/downloads). Because we cannot classify the ethnicities of each sample based only on the summary counts, we relied on the reported summary counts of Caucasian ancestry, which included 3093 cases and 8186 controls. The data were converted to the VCF format and annotated using ANNOVAR. We used the downloaded coverage bed file to extract high-coverage sites (at least 90% of samples with coverage ≥10) and intersected those with gnomAD high-coverage sites. In addition, we used the UCSC CRG Align 36 track to ensure that the mapping uniqueness was 1. The AF threshold was set to $5e{-}4$. We included variants annotated as stopgain, frameshift_insertion, frameshift_deletion, splicing, and non-synonymous with REVEL score ≥0.65. We applied our method to the case summary counts and summary counts from the gnomAD nfe population. There was a mild inflation in the QQ plot (Fig. 7a). Known genes *SOD1*, *NEK1*, and *TBK1* were ranked in the top. Two false positives were confirmed by manual variant QC check. For one false-positive gene, there were eight indels at the same position, chr12:53207583, and only homozygous genotypes were observed. The two false-positive genes had variants in both study-specific cases and controls but none in gnomAD. This observation was most likely due to processing platform-related batch effects. Because study-specific control summary counts were also available, we also performed the summary-count-based case–control rare variant analysis using case–control data only from this study. Yet, we still identified no obvious inflation in the QQ plot (Fig. 7b). For both cancer-predisposition genes *NEK1* and *TBK1*, using gnomAD showed a much-improved significance level, compared with that using the study-specific control summary counts. This demonstrates the advantage of using public

summary counts of large sample size in prioritizing disease-associated genes. Although minimal variant QC manipulation can be done with only summary counts available from cases and controls in this dataset, CoCoRV still successfully prioritized the disease-predisposition genes.

**The impact of ethnicity compositions on summary-count-based analysis**. Population structure is a known confounder for genetic association test if not properly addressed[8]. The effect of ethnic-specific allele frequencies (AFs) on the association test results depends on the differences among ethnic-specific AFs, and the differences in ancestry compositions between cases and controls. The majority population of gnomAD V2 exomes is European (nfe = 47%), and the four datasets we analyzed (ALL, CNS, GBM, LGG) all have European populations as the majority (nfe ≥70%, Supplementary Fig. S14). The dominance of European samples in the cases and controls, as well as our focus on rare variants (AF ≤ $1e{-}3$), might explain that in general, the results using ethnic-specific AFs or the pooled AFs did not show large differences in terms of genes achieving exome-wide significance for the four datasets we analyzed. However, we do observe one difference under the recessive model. Although LD was accounted for, using the pooled AF in analyzing the CNS cohort resulted in one false positive (*P* value <$2.5 \times 10^{-6}$), while there was no false positive if ethnicity-stratified analysis was used (Supplementary Data 10). The variant causing the false positive is 19-55898120-A-AT, which has the largest AF in gnomAD V2 African/African American (AF = 0.1122), but the AF is very low in European population (AF = 0.0002196).

To further illustrate the impact of ethnicity compositions and the importance of ethnic-specific analysis, we constructed a simulated "case" cohort using samples from the 1000 Genomes Project with sample ID starting with "NA". In total, there were 840 samples included from the five major populations (nfe: 210, afr: 250, amr: 65, eas: 213, sas: 102). The ethnicity composition in the constructed "case" was very different from that in gnomAD, e.g., the European samples were not the majority. Because the samples from the 1000 Genomes Project are not disease-specific, genes that were significant in the analysis between the constructed "case" and gnomAD were false positives due to the confounding of population structures, instead of being associated with any

disease. Association results showed that the simple pooled analysis can produce many false positives (Supplementary Fig. S15a, d and Supplementary Data 11), while ethnicity-stratified analysis substantially reduced the false positives (Supplementary Fig. S15c, f). Adding the filter using the maximal allele frequencies among gnomAD ethnicities (AF_popmax) could also mitigate the false positives (Supplementary Fig. S15b, e), although it was not as effective as ethnicity-stratified analyses. The variants that drive the top false positives in the simple pooled AF analysis showed large differences in allele frequencies between different ethnicities, where the AF ratios between the highest AF and lowest AF among ethnicities were around 10 and could reach about 20 (Supplementary Data 12). The number of false positives decreases as the AF threshold decreases. For example, when AF was set to $1e{-}4$ and the ethnicity-stratified analysis was used, there were no false positives in the analysis. The impact of different ethnicity compositions was not just limited to the African population but could happen in any other populations (e.g., eas, afr, amr, sas) less represented in the gnomAD dataset (Supplementary Data 12).

## Discussion

In this study, we developed a framework to prioritize disease-predisposition genes by using public summary counts as controls for rare variant burden tests. Consistent variant QC and filtering, as well as joint AF-based filtering make sure no artificial bias is introduced in the analysis. Ethnicity-stratified burden test ameliorates the effect of population stratification. By considering the discrete and heterogeneous nature of the null distribution, we proposed a more accurate inflation factor estimation method and several more powerful FDR-control methods than assuming the null distribution of $P$ values is uniform. We also proposed an approach to detect high-LD variants using gnomAD summary counts. We detected and stored the variants in LD from either using gnomAD summary counts or using in-house controls with full genotypes, and then used them in the framework to remove false positives, which is critical under the recessive models. In addition, we evaluated the concordance between using summary counts and full genotypes, which was high especially for the top hits with high statistical significance.

The inflation factor estimation and FDR-control method can also be used in other applications if the tests are performed using FET or CMH. Besides its application in the rare variant association test, our proposed LD test can detect high-LD MNVs, and is not limited by sequencing read length compared with traditional read-based MNV detection. Identifying rare variants in strong LD is also helpful in other analyses, e.g., distinguishing compound-heterozygous variants from those on the same haplotype or applying special treatment to MNVs[14].

In principle, allele frequencies derived from an independent large non-rare disease cohort should be helpful and can provide an unbiased way for AF-based filtering. We tried to use TOPMed AF (Freeze 8) annotations for variant filtering after lifted over to GRCh37 using picard[23]. Unfortunately using TOPMed AF alone introduced false positives (Supplementary Data 8). One main reason is that there are likely substantial differences in ethnicity proportions between gnomAD and TOPMed. Variants that show large ethnicity-specific AFs and should be filtered out are kept if we only use TOPMed's AF for filtering. One example is the variant 1-18809351-G–C, which has the highest AF in gnomAD V2 South Asian (AF = 0.1285), but TOPMed's overall AF is very low (AF = 0.00082360). Even though using TOPMed AF annotation alone for AF filtering can cause false positives, TOPMed could serve as a useful resource for additional AF-based filtering on top of the joint AF-based filtering.

By applying CoCoRV framework to pediatric and adult cancers, we identified known cancer-predisposition genes and prioritized other risk genes, one of which was statistically validated. We caution that CoCoRV should be used as a prioritization tool and not a statistical validation tool, because using summary counts might not control some hidden confounding factors. For example, our proposed method estimates the sample counts in the controls under different models rather than counting the actual counts from individual-level genotypes, there is the possibility that the estimated counts might not match well with the actual counts. This is the limitation of using summary-count-based burden tests. Once interesting genes are identified, either a strict full-genotype-based association test or lab-based functional studies are needed to validate the findings.

One potential limitation of using summary counts is that adjusting for covariates is not straightforward. Besides potential batch effects introduced by the sequencing platform, one (and often the only) confounding factor in genetic studies is population structure[24]. We propose to use a CMH-based, ethnicity-stratified analysis to mitigate this problem. Whether the population stratification causes an obvious systematic inflation depends on three factors: (1) population composition in the cases and controls, (2) AF threshold, and (3) sample size. In practice, we cannot guarantee that the populations of cases and controls will match perfectly, but with an adequately low AF threshold (e.g., 5e–4) and a focus on potential pathogenic variants, the influence of the population structure can be substantially reduced, possibly to the point of no systematic inflation. Improving the adjustment for population structure using summary counts might be worth future investigation.

Testing LD is a well-studied problem in statistical genetics when haplotypes or genotypes can be observed[25,26]. We extended the LD test when only a set of summary counts of two variants was available, such as in gnomAD. We assumed the Hardy–Weinberg equilibrium (HWE) in our test, which simplified the calculation and performed well. This might be due to that only strong LD can be well detected when using a limited set of summary counts, and the influence of the Hardy–Weinberg disequilibrium (HWD) for most rare variants is negligible or inadequate. Further investigation to extend the proposed method to account for the HWD might be useful, for example using a composite measure of LD[27]. Our proposed LD-detection method in CoCoRV is similar to a method proposed to estimate haplotype frequencies and LD measures in pooled DNA data[28]. The major differences are as follows: First, CoCoRV allows for different numbers of samples in a pool, e.g., the subset of samples within each ethnicity in gnomAD, rather than a fixed number in the design of the pooled experiment. Second, instead of using an expectation-maximization algorithm and treating the haplotype frequencies as missing data, our framework uses a direct gradient-based maximization of the likelihood, which exploits many well-developed gradient-based optimization methods[29]; therefore, it converges faster. In addition, we use a one-sided (rather than a two-sided) test to detect strong positive LD. Finally, we use the odds ratio (rather than correlation coefficient) to characterize LD strength for rare variants. In this study, we considered only the LD between two biallelic variants; in the previous study[28], multiple loci and multiple alleles were allowed, which might be an interesting topic to explore in the future.

One main difference between using the separately called summary counts and jointly called full genotypes was the variant QC step. For example, for the separately called summary counts-based analysis, VQSR was applied separately to cases and controls; however, for jointly called full-genotype-based analysis, it was applied to jointly called cases and controls. We found that the results of VQSR for a specific variant can differ between cases and

controls if run separately. This is partly expected because VQSR uses a machine learning approach to model the quality of variants within a jointly called genotype matrix. Different studies result in different training samples and different trained models. Including public sequencing data, such as the 1000 Genomes Project's WES data, with the case sequencing data will most likely improve the robustness of the variant QC and the concordance when the summary counts-based analysis is performed later.

The lack of obvious inflation in the QQ plot does not guarantee that the top hits are true positives; it indicates only that there is no large systematic inflation in the association test. However, sporadic individual false positives could be among the top hits. When using publicly available summary counts as controls, the cases and controls are processed from different pipelines using different parameters or software versions. Therefore, this approach is more prone to false positives than is the joint analysis of case and controls using full genotypes generated from the same pipeline and QC steps. Also, a more stringent check on the sequence region of the called variants would be helpful, such as alignment uniqueness, duplication segments, or repeats from the UCSC genome browser resources. Warning QC flags from gnomAD are also useful. The indel variants required more manual inspection due to challenges in the accurate alignment and QC. Another method that alleviates the false-positive problem is to process cases with a relatively smaller, publicly available sequencing dataset, such as the WES data from the 1000 Genomes Project, which is also advocated by GATK. Although this approach requires increased processing time and storage space, it helps separate the true enrichment of rare alleles in cases from the false enrichment due to pipeline differences.

Compared with the standard full-genotype-based analysis using the continuous PCs to account for population structure, the coarse assignment of discrete ethnicity groups cannot account for finer population structure within each ethnic group. Accurate inflation factor estimation and the QQ plot is thus critical for determining whether there is systematic inflation due to population structure. For ethnicity stratification, we used the 1000 Genomes Project's samples to predict the ethnicity of cases. This procedure was similar to the one used by gnomAD, though the final trained classifier was not the same due to a different training set. Hence, some discrepancies may occur between our assignment and those of the gnomAD classifier. We anticipate that the differences would be small, considering that the 1000 Genomes Project's samples are also part of the gnomAD dataset, and a very high probability threshold (0.9) was used to classify the samples in both our and gnomAD's ethnicity classification. Moreover, individual small discrepancies in the assignment are unlikely to substantially influence the final rare variant-based association test. When interpreting the association results, it is always more convincing if the association signals contributing to the final significance are found in more than one ethnicity.

In this study, both our processed data and gnomAD summary counts used GATK for variant calling and joint genotype calling, though there were differences, e.g., different versions, detailed implementations, and variant QC. By retaining high-quality variants and maintaining a consistent filtering strategy, we showed that batch effects in the processing pipeline can be well-controlled. However, we have not explored the potential batch effects if two completely different variant-calling algorithms were used (e.g., if one uses GATK and the other uses FreeBayes[30]). We anticipate these might introduce some inconsistencies and require further investigation. Recent standardization of some key processing steps in genome-sequencing analysis pipelines[31] shows promising results in minimizing these batch effects.

## Methods

**Data sources**. The case cohort in our main analyses included pediatric high-grade glioma (HGG) samples from the St. Jude PCGP, St. Jude LIFE study [central nervous system tumors (CNS) and acute lymphocytic leukemia (ALL)][32]. We constructed an in-house control cohort of samples using the WES data from the Alzheimer's Disease Sequencing Project (ADSP)[33] and the 1000 Genomes Project[34], which was later used to compare the concordance between using separately called control summary counts and jointly called case–control full genotypes. The datasets included multiple ethnicities (Supplementary Fig. S14), with European ancestry being the majority (52–81%). The inclusion of these individuals in our study was reviewed and approved by Institutional Review Board at St. Jude Children's Research Hospital. We also tested our framework in two other independent datasets, The Cancer Genome Atlas (TCGA) brain tumor cohort and an amyotrophic lateral sclerosis (ALS) study, to illustrate its power.

**Consistent quality control and filtering of variants**. When cases and controls are called separately, coverage summary information (e.g., the percentage of samples with coverage ≥10) ensures that regions of interest are well covered. Our tool incrementally summarizes coverage information and computes the percentage of samples with coverage no less than specified thresholds, such as {1, 5, 10, 15, 20, 25, 30, 50, 100}, for each nucleotide position. Our tool can scale well for tens of thousands of samples and is easy for parallel running (supplemental text). Following the coverage filtering in Guo et al.[4], we kept the variants that have ≥10 coverage (total reads of both alleles) in at least 90% of the samples for both cases and controls.

Inconsistencies can happen when QC and filtering are applied separately, e.g., when a variant is filtered out from controls but not from cases or vice versa. To address this problem, we employed the following strategy: keeping only high-quality variants from each cohort's QC process and performing consistent QC and filtering. For the former, all variants must pass the cohorts' QC filter. We required that the missingness within cases and controls be ≤0.1. In addition, we used the gnomAD data to generate a blacklist of variants based on gnomAD's filtering status and discrepancies in allele frequencies between the WES and WGS platforms (Supplemental methods). For consistent QC filtering, we excluded all variants that failed QC steps in either cohort. This consistent filtering step is absent in other tools such as TRAPD[4].

Another critical filtering step is joint allele frequency (AF)-based filtering. The joint AF of a variant is estimated by pooling the counts of cases and controls together, i.e., $AF = (AC_{case} + AC_{control})/(AN_{case} + AN_{control})$, where $AC_{case}$ and $AC_{control}$ are the alternate allele counts of cases and controls, respectively, and $AN_{case}$ and $AN_{control}$ are the total allele counts (including both reference alleles and alternate alleles) of cases and controls, respectively. Using joint AF for filtering avoids inconsistencies when separately filtering variants based on AF within cases and AF within controls.

The joint allele frequency (AF)-based filtering is consistent and unbiased, which is commonly applied when individual-level genotypes of cases and controls are available, such as in genome-wide association studies (GWAS). Suppose we filter cases using case-derived AF and filter controls using control-derived AF, it can cause inconsistencies because the case and control-derived AFs for the same variant can be quite different simply due to statistical variations which depend on the sample size of cases and controls. On the other hand, if only the control-derived AF is used for filtering, for controls, all variants selected will have AF less or equal to the specified AF threshold, but for cases, there are possibilities that some variants can have AF above the specified AF threshold simply due to statistical fluctuation. These inconsistencies can result in biased statistical test results and inflated type I error.

For ethnicity-stratified analysis, we first extract the AC and AN annotation from each ethnicity, and then calculate the ethnicity-specific joint AFs, then the maximal joint AF among all ethnicities is used for the AF-based filtering. This shares some similarities of using the maximal gnomAD AFs (AF_popmax) among all ethnicities for filtering, however, our filtering is based on the joint AF considering both cases and controls instead of the control-derived (gnomAD-derived) AF.

**Estimation of sample counts in three models using summary genotype counts**. We defined three models to count the samples for the burden test (Fig. 1). Let $AC_i$ be the alternate allele count of each variant $i$ in a gene for a sample. A sample belongs to one or more of the following models if the sample satisfies the corresponding conditions:

$$
\begin{cases}
\text{dominant model}: \sum_i AC_i \geq 1 \\
\text{recessive model}: \sum_i AC_i \geq 2 \\
\text{double-heterozygous model: at least two variants with } AC_i \geq 1
\end{cases}
\tag{1}
$$

Here the recessive model can be either homozygous with alternate alleles at the same position, or at least one alternate allele at ≥2 positions, i.e., double-heterozygous. Therefore, the double-heterozygous model can be viewed as a special case of the recessive model. As noted in Guo et al.[4], because the haplotype information is not directly observed, the double-heterozygous model could mean two variants on the same haplotype, thus not exactly a compound-heterozygous

model. We often have the full genotypes for cases, so we can determine the count of each model directly from the genotype matrix. However, for controls, we need to estimate the number of samples qualified for each model based on the summary counts of each variant. Specifically, for a gene with $m$ variants, let $p_{iG}$ be the genotype frequency of variant $i$ with genotype $G$, where $i = 1, \cdots, m$, and $G = 0, 1, 2$. Public summary counts, such as gnomAD, usually provide this information; however, if only AF is available, the genotype frequencies must be estimated. One convenient, though not optimal, approach is to assume the Hardy–Weinberg equilibrium (HWE). We estimate the probability of a sample satisfying the defined models in (1) as follows. Denote the probability of the dominant model by $p_{DOM}$, the recessive model by $p_{REC}$, and the double-heterozygous model by $p_{2HET}$. If only one variant exists, then $p_{DOM} = 1 - p_{10}, p_{REC} = p_{12}$, and $p_{2HET} = 0$. If there are at least two variants, they can be estimated as shown below, assuming independence between the rare variants:

$$p_{DOM} = 1 - \prod_{i=1}^{m} p_{i0} \qquad (2)$$

$$p_{2HET} = \sum_{i=1}^{m-1} \sum_{j=i+1}^{m} \sum_{k \neq i, k \neq j} \left(1 - p_{i0}\right)\left(1 - p_{j0}\right) p_{k0} \qquad (3)$$

$$p_{REC} = \sum_{i=1}^{m} p_{i2} \sum_{j \neq i} p_{j0} + p_{2HET} \qquad (4)$$

Equation (2) calculates the probability of being dominant for at least one locus. For (3), we approximated the probability by considering all pairs of variants with AC ≥1 out of $m$ loci, because the probability of having two variants with AC ≥1 is much higher than that of having three or more variants with AC ≥1. Similarly, in (4), we considered a single variant with homozygous alternate alleles and two variants with AC ≥1. To calculate the estimated counts, we multiplied the frequencies of each model by the total number of controls. Here we assumed all rare variants were independent; later we made some modifications when we detected variants in LD.

**Burden test with samples stratified by ethnicity groups.** In addition to the pooled analysis of all ethnicities, we provide an ethnicity-stratified analysis. The latter requires predicting the ethnicity of each case. We performed principal component (PC) analysis for all samples, including samples from the 1000 Genomes Project (see the supplemental text). Following the practice in gnomAD[1], we sampled a random forest classifier based on the top 10 PCs and used the six major ethnicity categories as training labels: non-Finnish European (nfe), African American (afr), Admixed American (amr), East Asian (eas), South Asian (sas), and Finnish from the 1000 Genomes Project. Then we applied the classifier to the rest of the samples and used probability 0.9 as a cutoff to assign samples to the six ethnicities or to "other" when all predicted probabilities were less than 0.9. We then stratified samples based on the six ethnicities and calculated/estimated the counts for each model within each stratum; samples labeled "other" were not used. We performed the CMH-exact test[35] to aggregate evidence of each stratified 2 × 2 contingency table. When using gnomAD summary counts as controls, we used the stratified summary counts of each ethnicity directly.

**Inflation factor estimation for discrete count-based test.** For the $P$ values of burden tests calculated by the FET or CMH, the null distribution does not follow the uniform distribution. Therefore, the expected ordered null $P$ values could not be derived from the uniform distribution. The null distribution of the ordered $P$ values of a set of tested genes using either FET or CMH depends on the number of rare alleles in each gene. Instead of assuming the incorrect uniform distribution, we empirically sampled from the correct null distributions for inflation factor estimation. We assumed that the genes are independent of each other, which is reasonable for rare variants with very low AFs. Instead of permuting the phenotypes, we worked on the 2 × 2 contingency table directly (Fig. 2a). For each gene, let $m$ be the number of cases, $n$ be the number of controls, and $k$ be the number of samples with rare alleles among all cases and controls. These values can be obtained from the observed 2 × 2 contingency tables and are held fixed for each gene. In order to sample $P$ values under the null hypothesis of no association, for each gene, we can randomly sample the number of cases with rare alleles (denoted by $x$) from a hypergeometric distribution with parameters $m$, $n$, $k$ and form a new 2 × 2 contingency table, then we can calculate the null $P$ values using FET. However, this strategy will need to run FET for each replicate, which can be time consuming for all the genes and all replicates. A much faster sampling of $P$ values under the null is to sample directly from the $P$ values' cumulative distribution function (CDF). To generate the CDF, first we generated the $P$ values' support list which is based on tabulating all possible values of $x$ and corresponding probabilities from the hypergeometric distribution. Then the $P$ value can be computed for each value of $x$ according to the hypothesis and CDF can be generated afterward. We used the CDF approach for fast sampling. Then we sorted the $P$ values under null across all genes for each replicate. This process was repeated $N$ times, and the final expected sorted $P$ values (order statistics) were the average of $N$ $P$ values at each ordered rank. To estimate the inflation factor, we took the lower 95% quantile of points in

the QQ plot and regressed the sorted $\log_{10}$-scaled observed $P$ values to the $\log_{10}$-scaled expected sorted $P$ values. The slope of the regression was used as the inflation factor, which is denoted by $\lambda_{emp}$. Similarly, we extended the inflation factor estimation to the CMH-exact test. For CMH-exact test[36], assume there are $K$ strata, the test statistic is $S = \sum_{i=1}^{i=K} x_i$, where $x_i$ is the number of case samples with rare alleles in strata $i$. To sample the $P$ values directly from the CDF under the null hypothesis, we first need to tabulate all possible values of $S$ and corresponding probabilities. Direct enumeration of $S$ would be very time consuming, fortunately, there has been a very fast network-based algorithm developed[37]. We use the network-based algorithm, which was already implemented as an internal function in R, to calculate the $P$ values' support. Then the $P$ value can be computed for each value of $S$ according to the null hypothesis and the CDF can be generated afterward. Once the $P$ values under the null are sampled based on the CDF, the inflation factor could then be calculated similarly as in FET.

Guo et al.[4] also noted an excess of $P$ values that equaled 1 when the null distribution was assumed to follow a uniform distribution. Thus, they proposed modifications to estimate the inflation factor. Two versions of this estimation are included in Guo et al.[4]. One estimated the slope by using values between quantile 0.5 and 0.95, denoted by $\lambda_{0.5-0.95}$ in our study, and the other estimated the slope from two points: the first is the point with the observed $P$ value of 1 with the highest rank, and the second is the point at the 0.95 quantile of the $P$ values not equal to 1, denoted by $\lambda_{2\text{points}}$ in our study. For both $\lambda_{0.5-0.95}$ and $\lambda_{2\text{points}}$, the expected ordered $P$ value of rank $r$ in an increasing order from $n$ $P$ values was $r/(n + 1)$ assuming a uniform distribution $U(0, 1)$. Because these modifications were still based on a uniform distribution, their solutions were biased.

To illustrate the performance of different methods on inflation factor estimation, we used simulated $P$ values under the null hypothesis to evaluate the bias. Specifically, for $N$ replicates of sampled null $P$ values, we applied all three methods to estimate the inflation factor. For the empirical null $P$ value-based method, we saved on computation cost for each replicate of null $P$ values by using the rest $(N−1)$ replicates of null $P$ values to estimate the expected sorted $P$ values.

**FDR control.** Resampling-based FDR-control method was originally developed to address the correlation between multiple tests[38]. We adopt it here for FDR control of discrete count-based tests. We simulated the $P$ values under the null hypothesis the same as in the inflation factor estimation (Fig. 2a). Then we adopted the resampling-based FDR-control method to estimate the adjusted $P$ values. The two resampling methods used a mean point estimate (RBH_P) or an upper limit estimate (RBH_UL) for the number of true positives detected, respectively. Given the simulated $P$ values under the null hypothesis, the adjusted $P$ value estimation is similar as that in the R package FDR-AME[39]. Let $R^*(p)$ be the number of genes with $P$ values no greater than $p$ in the simulated $P$ values under the null hypothesis, $R(p)$ the number of genes with $P$ values no greater than $p$ in the observed $P$ values, $M^*(p)$ the mean of $R^*(p)$, $Q_\beta^*(p)$ the $1−\beta$ quantile of $R^*(p)$. Then for RBH_P, the estimated true positives detected at threshold $p$ was $s(p) = R(p) - M^*(p)$; for RBH_UL, $s(p) = R(p) - Q_\beta^*(p)$. Then the adjusted $P$ values were calculated as

$$\text{PBH\_P}(p) = \begin{cases} E_{R^*}\left(\frac{R^*(p)}{R^*(p)+s(p)}\right), & \text{if } s(p) \geq Q_\beta^*(p) \\ E_{R^*}\left(R^*(p) > 0\right), & \text{otherwise} \end{cases} \qquad (5)$$

and

$$\text{PBH\_UL}(p) = \begin{cases} E_{R^*}\left(\frac{R^*(p)}{R^*(p)+s(p)}\right), & \text{if } s(p) > 0 \\ E_{R^*}\left(R^*(p) > 0\right), & \text{otherwise} \end{cases} \qquad (6)$$

where $E_{R^*}$ means taking the expectation over all replicates of $P$ values under the null hypothesis.

An alternative approach for powerful FDR control is using FDR methods which directly account for the discreteness and heterogeneity of the distribution. We chose to use the methods developed recently which show good performance for discrete distributions[40], e.g., A-DBH-SD. These methods can be applied to arbitrary discrete distributions if the CDF of the $P$ values under the null is known or can be calculated. However, the implemented R package DiscreteFDR only supports the generation of CDF of FET. To extend these powerful FDR methods to the CMH-exact test, we developed a method to generate the CDF of the $P$ values under the null in CMH-exact test as described in the section for inflation factor estimation. Due to the fast network algorithm[37] for tabulating the test statistic and corresponding probabilities, the CDF list can be generated within seconds for ~20,000 genes considered.

**Detection of high-LD rare variants via gnomAD summary counts.** One interesting feature of CoCoRV is the proposed LD test using only gnomAD summary counts (Fig. 2). The gnomAD dataset has several subset-based summary counts, including "controls" and "non_cancer." Because control samples are a subset of non_cancer samples, we can partition them into three independent sets of summary counts: cancer, healthy controls within the non_cancer set, and other diseases within the non_cancer set, which are labeled "non_cancer_non_controls." After further stratifying by sex, we can generate six independent summary counts per

ethnicity (Fig. 2). We assume that the allele frequencies and LD among variants are the same among these six independent datasets per ethnicity, which is likely reasonable because most variants should not be associated with sex or cancer. Given the $ACs$ (alternate allele counts in a cohort) of two variants and the total number of haplotypes in these data, we can test the hypothesis that two variants are in positive LD, i.e., the variants are more likely to lie on the same haplotype than random chance under the assumption of independence. Specifically, let the data observed be $\{x_i, y_i, n_i, i = 1 \cdots I\}$, where $x_i$ is the $AC$ of the first variant, $y_i$ is $AC$ of the second variant, $n_i$ is the total number of haplotypes, $I$ is the number of independent sets of summary counts (e.g., six from gnomAD) (Fig. 2). Denote the four haplotypes of two variants by $h_{11}, h_{10}, h_{01}, h_{00}$ and their corresponding probabilities by $p_{11}, p_{10}, p_{01}, p_{00}$, where 1 indicates the alternate allele and 0 indicates the reference allele. The log-likelihood of the observed data is

$$\log L = \sum_{i=1}^{I} \log P(x_i, y_i | n_i) \qquad (7)$$

In the following equation, we drop the subscript $i$ to simplify the notation. Let $r$ be the count of unobserved haplotype $h_{11}$, then the counts of the four haplotypes are $r, x - r, y - r$, and $n - x - y + r$, respectively. By the law of total probability, the likelihood of observed allele counts $x, y$ of two variants given $n$ total haplotypes is $P(x, y | n) = \sum_r P(x, y, r | n)$. Assuming the HWE, $P(x, y, r | n)$ can be calculated as the multinomial probability with the four haplotype counts and probabilities. Specifically,

$$P(x, y, r | n) = \frac{n!}{r!(x-r)!(y-r)!(n-x-y+r)!} p_{11}^r p_{10}^{x-r} p_{01}^{y-r} p_{00}^{n-x-y+r} \qquad (8)$$

Note that the range of $r$ is $[\max(0, x + y - n), \min(x, y)]$. There are only three free parameters for the four haplotype probabilities because their sum is 1. For a direct test of the null hypothesis, we reparametrize the haplotype probabilities using another three parameters $s = p_{11} + p_{10}$, $t = p_{11} + p_{01}$, and $\theta = \frac{p_{11}p_{00}}{p_{10}p_{01}}$, where $s$ is the alternate AF of the first variant, $t$ is the alternate allele frequency of the second variant, and $\theta$ is the odds ratio specifying the LD strength and direction between two variants. We used a likelihood ratio test, where the null hypothesis was $\theta \leq 1$ and the alternative hypothesis was $\theta > 1$, i.e., the two variants were more likely to be on the same haplotype. The parameters $s$ and $t$ were treated as nuisance parameters. Specifically, the test statistic is as follows:

$$T = 2 \left( \max_{s, t, \theta \in (-, +)} \log L(s, t, \theta) - \max_{s, t, \theta \in (-, 1)} \log L(s, t, \theta) \right)$$

We derived the gradient of the log-likelihood function and used the R package *nloptr*[29] to maximize the log-likelihood under the null hypothesis and full model. For robust maximization, multiple initial parameter values were used. The chi-square distribution with 1 degree of freedom was used to calculate the $P$ values, which appeared to work well, though the asymptotic distribution of the one-sided likelihood ratio test could be better characterized. We used an odds ratio of 1 in the null hypothesis; however, other prespecified odds ratios can be used to test directly whether the LD exceeded a high odds ratio threshold. Due to QC, the $n_i$ of each variant can slightly differ, hence in our implementation, it was calculated as the rounded average of all $(n_i)$s of the qualified variants within a gene. To accelerate the optimization, we implemented the gradient functions in C++ with Rcpp[41].

The above test can also be used to detect LD when full genotypes are observed. The additive coding of genotypes corresponds to $n = 2$ in the above summary-count-based test. The calculation of the likelihood can be accelerated because there are only nine combinations of genotypes between the two variants. Let $f_{ij} = P(x = i, y = j | n = 2)$, then assuming HWE we have

$$f_{00} = p_{00}^2; f_{01} = 2p_{00}p_{01}; f_{02} = p_{01}^2;$$

$$f_{10} = 2p_{10}p_{00}; f_{11} = 2p_{11}p_{00} + 2p_{10}p_{01}; f_{12} = 2p_{11}p_{01}; \qquad (10)$$

$$f_{20} = p_{10}^2; f_{21} = 2p_{11}p_{10}; f_{22} = p_{11}^2.$$

The log-likelihood of the data is as follows:

$$\log L = \sum_{i=0}^{2} \sum_{j=0}^{2} c_{ij} \log(f_{ij}) \qquad (11)$$

where $c_{ij}$ is the count of genotype combination $x = i, y = j$. This full-genotype-based LD test resembled a recently proposed LD test[26], but we used a different parametrization. We also implemented this full-genotype-based LD test in CoCoRV, in case users have full genotypes, which will result in higher power to detect LDs.

**Accounting for LD between two variants**. For variants in high LD, we needed to adjust the procedure of counting qualified samples with rare alleles. When estimating the counts of qualified samples in controls for each group of rare variants in LD, only the variant with the highest AF was kept representing the variants in LD; the rest were excluded. The reason that we exclude redundant variant pairs that are in LD in the controls from the counting of the recessive model is twofold: (1) to match the assumption when estimating the number of samples under the recessive models in controls: rare variants are assumed to be independent; (2) it is less interesting if the recessive pattern in the case samples can be simply explained by LD between two variants in the control population. Supplementary Fig. S16

illustrates the counting process under different models with/without considering LD between pairs of rare variants in LD.

For a quick check of high-LD variants, we precomputed the LD test results ($P$ values and FDRs) from gnomAD and in-house controls and stored those with relatively small $P$ values (e.g., $P < 0.1$). When estimating sample counts for controls, we regarded variants as LD variants if the LD test result has FDR < 0.05. To be conservative, for each double-heterozygous sample in the cases, we further required the $P$ value of the LD test to be less than a threshold (e.g., $P < 0.05$ or 0.1). This practice reduced false positives due to strong LD between variants under the recessive or double-heterozygous models.

**Detection of high-LD variants in gnomAD exomes**. We used our proposed LD-detection method to scan the gnomAD exome-based summary counts in each gene to detect high-LD variants, which share some similarities with MNV[14]. MNVs are usually detected by direct examination of the sequence reads in aligned bam files. As a validation of the effectiveness of the proposed test, we detected high-LD variants by using the summary counts from the gnomAD exomes and compared them with the reported gnomAD MNVs.

Because we are interested in rare variants annotated with some functions, we focused on variant sets annotated as "stop gain", "nonsynonymous", "splicing", "frameshift_insertion", or "frameshift_deletion" by ANNOVAR. We restricted variants in the cohort to those with AF ≤0.01, at least 10 alternate allele counts ($AC$ ≥ 10), and missingness ≤0.1. Note that the alternate allele count differed from the alternate read count, which is the number of sequencing reads harboring the alternate allele supporting the genotype calls. We filtered out variants in the blacklist, as described in the supplemental text. We required a coverage depth of at least 10 (total reads of both alleles) for 90% of the samples. For each pair of variants within a gene, we applied the LD test by using the summary counts for each ethnicity. In total, about 10 million tests were run. Variant pairs with FDR < 0.05 within each ethnicity were considered significant. For comparison with gnomAD MNVs, we downloaded the coding MNVs detected from gnomAD exomes, where variants' distances were ≤2 base pairs (BPs). Therefore, we restricted our comparison to high-LD variants with distances ≤2 BPs. We also downloaded the MNV lists based on gnomAD genomes and defined the union of coding MNVs and genome MNVs with distances ≤2 BPs as the full set of MNVs reported in gnomAD with ≤2 BPs.

**Comparison of CoCoRV with other methods**. TRAPD[4] was developed to use gnomAD summary counts, therefore we perform the comparison with CoCoRV using gnomAD summary counts. The schematic diagram of applying TRAPD is shown in Supplementary Fig. S17. TRAPD proposed to use a single annotation QD for variant QC. We followed the description in TRAPD[4] to select the best QD threshold pairs for cases and controls by which TRAPD's estimated inflation factor was closest to 1. Besides the QD-based filtering, we also tried the option "--pass" for variant filtering which is based on the FILTER status. Because the filtering is performed in cases and controls separately, it can create inconsistencies. We used "--minAN" to make sure the variant missingness ≤0.1, which can help reduce false positives. The option "--popmaxAF" is used for filtering based on AF_popmax in gnomAD. We use TRAPD's "--includeinfo" to select variants based on the defined criteria of potential pathogenic variants. Because TRAPD uses the maximal allele frequency among different ethnicities (AF_popmax) to filter variants, we added AF_popmax to filter variants in CoCoRV. Specifically, we used an AF_popmax threshold of 1$e$−3 in TRAPD for filtering in the final test with the best QD threshold for single-nucleotide variants (SNVs) and indels. For CoCoRV, we used an AF threshold of 1$e$−3 and AF_popmax threshold of 1$e$−3. Because TRAPD uses only $AC$ and $AN$ to derive qualified sample counts, we included the analysis using $AC$ and $AN$ from gnomAD for CoCoRV. In addition, we included the ethnicity-stratified results for CoCoRV. We changed the default one-sided FET in TRAPD to a two-sided FET to match that used in CoCoRV.

Another method ProxECAT[16] which pools the counts of functional and non-functional alleles between cases and controls to form a 2 × 2 contingency table for the test of the enrichment of rare variants. However, the R package ProxECAT only provides two functions performing the statistical tests assuming the contingency tables have already been generated. There is no tool for any upstream filtering or generation of the 2 × 2 contingency tables. For a fair comparison, we implemented the pooling method proposed by ProxECAT in CoCoRV, and used the same upstream variant filtering from CoCoRV. The functional alleles were defined the same as that in CoCoRV for potential pathogenic alleles. The non-functional alleles included alleles annotated as synonymous or missense variants with REVEL < 0.2. The test in ProxECAT is essentially the likelihood ratio test given the 2 × 2 contingency table, which might not produce accurate $P$ values when the counts are very small, therefore we replace it with Fisher's exact test in our analysis.

**Reporting summary**. Further information on research design is available in the Nature Research Reporting Summary linked to this article.

# Data availability

GRCh37-lite is available at ftp://ftp.ncbi.nih.gov/genomes/archive/old_genbank/ Eukaryotes/vertebrates_mammals/Homo_sapiens/GRCh37/special_requests/GRCh37-

lite.fa.gz. UCSC CRG Align 36 track is available at http://hgdownload.cse.ucsc.edu/goldenpath/hg19/encodeDCC/wgEncodeMapability/wgEncodeCrgMapabilityAlign36mer.bigWig. The gnomAD summary-count data is available at https://gnomad.broadinstitute.org/. The gnomAD detected MNV is available at https://gnomad.broadinstitute.org/downloads#v2-multi-nucleotide-variants. The TOPMed summary-count data is available at https://bravo.sph.umich.edu/. The ALS data used in this study are available at http://alsdb.org/downloads. The Alzheimer's Disease Sequencing Project (ADSP) are available under restricted access following NIH dbGaP's policy: https://www.ncbi.nlm.nih.gov/projects/gap/cgi-bin/study.cgi?study_id=phs000572.v8.p4. The 1000 Genomes Project's data are available at https://www.internationalgenome.org/. The Cancer Genome Atlas (TCGA) is available under restricted access following NIH dbGaP's policy: https://www.ncbi.nlm.nih.gov/projects/gap/cgi-bin/study.cgi?study_id=phs000178.v11.p8. The St Jude cancer cohort is available under restricted access following St Jude Cloud's policy: https://www.stjude.cloud/. The GTEx portal is available at https://gtexportal.org/home.

## Code availability

The code for TRAPD is available at https://github.com/mhguo1/TRAPD. The code for ProxECAT is available at https://github.com/hendriau/ProxECAT. The CoCoRV[42] code for summary counts-based rare variant association test is available at https://bitbucket.org/Wenan/cocorv/src/master/.

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

## Acknowledgements

This study was supported in part by the National Cancer Institute grant P30 CA021765 [G.W.] and 5U54NS092091-08 [G.W.], American Lebanese Syrian Associated Charities (ALSAC), and the U.S. Public Health Service and National Institutes of Health (contract grant number R35 GM140487) [D.J.S.]. The content is solely the responsibility of the authors and does not necessarily represent the official views of the National Institutes of Health. We acknowledge the permission granted to use data from the Alzheimer's Disease Sequencing Project (dbGaP Study Accession phs000572.v8.p4) and The Cancer Genome Atlas (dbGaP Study Accession phs000178.v11.p8). We acknowledge Michael Edmonson's help with the conversion between plain text-based matrix data format and the VCF format, Jason P. Sinnwell's help on LD packages, and Angela J. McArthur's helpful scientific editing.

## Author contributions

W.C. proposed the study design, developed methods, implemented methods, processed the data and performed data analysis, reviewed the results, and drafted the manuscript. S.W. implemented methods and processed the data. S.S.T. implemented methods and processed the data. D.W.E. provided guidance on phenotype subtype information and reviewed the results. D.J.S. reviewed the draft and provided guidance on method development, G.W. proposed the study design, processed the data, reviewed the results, and drafted the manuscript. All authors read and approved the final manuscript.

## Competing interests

The authors declare no competing interests.
