## [Peer Review File · Nature Communications]

A rare variant analysis framework using public genotype summary counts to prioritize disease-predisposition genesReviewers' Comments:

Reviewer #1:

Remarks to the Author:

A huge volume of sequencing data have been generated through significant efforts from large sequencing projects such as gnomAD. The utilization of these publically available data for in-house projects remains a great challenge due to a number of factors such as different sequencing platforms, data analysis pipeline and parameters, and batch effects. It's infeasible to re-analyze from raw sequencing data to address the challenges in most research groups.

In this study, Chen and colleagues have developed an analytical framework, consistent summary counts based rare variant burden test (CoCoRV), aiming to maximize the use of data (i.e. summary counts of variants) to facilitate genetic association studies. They applied several filtering steps and data quality controls in order to produce high-quality summary counts for downstream analyses. The CoCoRV framework includes some additional features, such as ethnicity-stratified rare variant association test, estimation of inflation factors, FDR control, and rare variants in high linkage disequilibrium. Overall, this is well-motivated and conducted study aiming to providing a cost-effective solution for the use of public sequencing data in genetic association studies. I have some specific comments to be addressed about this work below.

1) In the filtering steps, the authors analyzed variants that have ≥ 10 coverage in at least 90% of the samples for both cases and controls. It has been known sequencing coverage is the most critical factor for the accuracy of variant calling. The authors should perform further analyses to clarify impacts of using this cutoff (i.e. how about ≥ 20 , ≥ 50 or other coverage). It's unclear if ≥ 10 coverage was applied to alternative allele or both alleles?

For the joint allele frequency (AF)-based filtering step, they need to provide more details about this. I believe this step could significantly affect the downstream association analysis.

2) For rare variants in high linkage disequilibrium analysis, they proposed a statistical approach using log likelihood of the observed data from six independent sets (i.e., stratified by Sex). It will be helpful if they can use individual-level data to demonstrate their accuracy (the reliability is questionable due to the rare property), given they have such in-house large-scale individual-level data. As LD information between variants can be easily derived from the individual data, this approach component could be not necessary (especially a concern here).

3) When they excluded variants in LDs for the association analysis under the CoCoRV framework, it's unclear why those top associations are false positives (in Fig. 5). It's also unclear why all these association signals for variants in LD were removed? Should remain one association signal from the most possible causative variant?

4) In Method section of Estimation of sample counts in three models using summary genotype counts. For formula (2)-(4), the authors assumed independence between the rare variants. It's unclear how to calculate a burden test for a gene with multiple variants if they don't have individual data (no information to tell if a subject carry alternative alleles of multiple variants for a gene of interest). This could be a limitation to be discussed for the gene-based association analysis.

Reviewer #2:

Remarks to the Author:

The authors report a refinement and improvement in a previously published method to find genes implicated in rare diseases. More information is needed to understand how this updated method change is the scientific conclusions drawn from this type of analysis.

It would be helpful for each analysis to:

1) Annotate with allele frequency using neither the case nor the control data set. For example, the use of TOPMed could be considered.

2) Uniformly report results when comparing between two methods. For example in this assessment X

number of genes using Y program and the C number of genes using B program achieved exome wide significance. Of those that achieve this level of significance, Y program generated xx% of false positives and B program generated XX percent of false positives. These false positives were determined to be due to the following issues. I cannot tell in the results if the authors are reporting on false positives for signals that didn't cross an exome-wide significant threshold.

3) there are several programs that allow for this type of rare variant burden testing, is there a reason why the authors only chose to compare to TRAPD?

4) The authors allow for consideration of ethnic specific minor allele frequencies —How much did this really change the results? Was this impact limited to African populations?

5) TRAPD allows you to annotate and select variants based on a variety of metrics. Is there a reason why the authors are limiting it to a single metric?

Response to reviewers' comments

We appreciate both reviewer's insightful comments about our study. These comments greatly improved both the quality of the paper and our developed tool. Below are our responses to reviewer's comments point by point. We have also highlighted changes in the manuscript in blue.

Reviewer #1 (Remarks to the Author):

A huge volume of sequencing data have been generated through significant efforts from large sequencing projects such as gnomAD. The utilization of these publically available data for in-house projects remains a great change due to a number of factors such as different sequencing platforms, data analysis pipeline and parameters, and batch effects. It's infeasible to re-analyze from raw sequencing data to address the challenges in most research groups.

In this study, Chen and colleagues have developed an analytical framework, consistent summary counts based rare variant burden test (CoCoRV), aiming to maximize the use of data (i.e. summary counts of variants) to facilitate genetic association studies. They applied several filtering steps and data quality controls in order to produce high-quality summary counts for downstream analyses. The CoCoRV framework includes some additional features, such as ethnicity-stratified rare variant association test, estimation of inflation factors, FDR control, and rare variants in high linkage disequilibrium. Overall, this is well-motivated and conducted study aiming to providing a cost-effective solution for the use of public sequencing data in genetic association studies. I have some specific comments to be addressed about this work below.

Response:

Thank you for recognizing our efforts in this study. Since the initial submission, we have further improved the speed of the estimation of inflation factors by sampling directly from p-values' cumulative distribution function (CDF) under the null hypothesis of no association (revised Figure 2a). The computing only takes less than one minute for sampling 1,000 replications across ~20,000 genes. For FDR control, in addition to the powerful resampling based FDR method, we also extended the FDR method designed directly for discrete and heterogeneous distributions from Fisher's exact test to the CMH exact test (revised Figure 2a, Section "Inflation factor estimation for discrete count based test & FDR control"). Moreover, for detecting rare variants in high linkage disequilibrium, we have incorporated the LD detection results from using individual level genotypes of the in-house constructed synthetic controls (n=8,175), thanks to your comments about LD detection using individual level genotypes. More details are discussed in the following response.

1) In the filtering steps, the authors analyzed variants that have ≥ 10 coverage in at least 90% of the samples for both cases and controls. It has been known sequencing coverage is the most critical factor for the accuracy of variant calling. The authors should perform further analyses to clarify impacts of using this cutoff (i.e. how about ≥ 20 , ≥ 50 or other coverage). It's unclear if ≥ 10 coverage was applied to alternative allele or both alleles?

Response:

Thank you for pointing this critical factor out. We applied the coverage cutoff ≥ 10 to the total reads of both alleles, instead of alternative allele, which have been clarified in the section: “Consistent quality control and filtering of variants”.

Per your suggestions, we have evaluated the impact of different coverage thresholds ≥ 1 , ≥ 5 , ≥ 10 , ≥ 15 , ≥ 20 by calculating the concordance between using separately called summary counts and the jointly called full genotype based case-control data, adding the following analysis results in the section: “Concordance of rare variant association between using summary counts and full genotypes”.

“The coverage depth cutoff of 10 shows the best concordance measured by correlations of qualified sample counts in cases or controls, and correlations of the association test p-values (Table S2). The best concordance of using the depth cutoff 10 could be related to the QC of genotypes: we keep high quality genotypes with $DP \geq 10$ (DP is the number of informative reads for each sample in the VCF file), which is consistent with the QC used in gnomAD for summary allele counts. As the coverage cutoff increases, the variant quality will increase too, but we might risk missing good quality variants if the coverage cutoff is too high. We also calculated the size of qualified regions with different coverage cutoffs on gnomAD’s whole exome coverage data (Fig. S3). The coverage cutoff of 10 seems to be a good trade-off between variant quality and the size of qualified regions being retained.”

For the joint allele frequency (AF)-based filtering step, they need to provide more details about this. I believe this step could significantly affect the downstream association analysis.

Response:

We agree with you that this filtering step is important in order to achieve consistent and unbiased filtering. We have added more details as below in the section: “Consistent quality control and filtering of variants”.

“The joint allele frequency (AF)-based filtering is consistent and unbiased, which is commonly applied when individual level genotypes of cases and controls are available, such as in genome-wide association studies (GWAS). Suppose we filter cases using case-derived AF and filter controls using control-derived AF, it can cause inconsistencies because the case and control-derived AFs for the same variant can be quite different simply due to statistical variations which depend on the sample size of cases and controls. On the other hand, if only the control-derived AF is used for filtering, for controls, all variants selected will have AF less or equal to the specified AF threshold, but for cases, there are possibilities that some variants can have AF above the specified AF threshold simply due to statistical fluctuation. These inconsistencies can result in biased statistical test results and inflated type I error.

For ethnicity stratified analysis, we first extract the AC and AN annotation from each ethnicity, and then calculate the ethnicity specific joint AFs as described above, then the maximal joint AF among all ethnicities is used for the AF based filtering. This shares some similarities of using the

maximal gnomAD AFs (AF_popmax) among all ethnicities for filtering, however, our filtering is based on the joint AF considering both cases and controls instead of the control-derived (gnomAD-derived) AF.”

2a) For rare variants in high linkage disequilibrium analysis, they proposed a statistical approach using log likelihood of the observed data from six independent sets (i.e., stratified by Sex). It will be helpful if they can use individual-level data to demonstrate their accuracy (the reliability is questionable due to the rare property), given they have such in-house large-scale individual-level data.

Response:

Thank you very much for the suggestion. We followed your suggestion to validate the detected LD pairs from gnomAD using our in-house individual level full genotype as well, with additional improvement in LD detection.

In particular, we have first updated our implementation to make the proposed LD test more robust by using multiple initial seed values in the maximization of the likelihood, and adding the r^2 and D prime metric output besides the odds ratio to describe the LD strength. After the update, there are a few minor count changes in Table 2 and a few LD-pair changes in Table S5, without changing any of our analysis conclusions. To check the reliability, we compared our tool with the *ld* function in *snpStats*¹ in the setting of full genotypes, which is a special case of our general method. Then we did the validation using the large controls we constructed in-house (n=8,175). We added the following in the section: “The proposed LD detection method has high power in detecting high-LD rare variants and accurately identifies MNVs in gnomAD”

“... because the full genotype based LD test is a special case of our proposed method, where the total haplotypes is 2 for all individuals (groups). We evaluated our method in simulations under the full genotype setting and compared it with the ld function in snpStats¹ which is designed for LD detection using full genotypes. CoCoRV and snpStats have almost the same results (Fig. S5b) and the type I error was well controlled. ...

To further demonstrate the accuracy of detected LD pairs from gnomAD (FDR < 0.05 within each ethnicity), we validated them using the constructed 8,175 controls with individual-level genotypes. Specifically, we used the ld function in the R package snpStats¹ to test each LD pairs within the corresponding ethnicities. Out of 10,081 LD pairs from gnomAD, 7,290 (72.3%) variant pairs can be tested with at least one alternate allele from at least one variant of the pair in the individual-level controls. We used a p-value threshold 5×10^{-4} , which corresponds to 3.6 expected false positives out of 7,290 tests under the null hypothesis of no LD. Of the 7,290 variant pairs tested, 6,989 (95.9%) passed the p-value threshold 5×10^{-4} (Table S5). All pairs that passed the threshold except three have odds ratio larger than 150. These results show that our proposed method of detecting LD from gnomAD using summary counts is accurate or has high precision.”

2b) As LD information between variants can be easily derived from the individual data, this approach component could be not necessary (especially a concern here).

Response:

Thank you very much for bringing this up. Although LD can be detected when there is individual data, the studied case cohort usually is small, preventing detecting the LD for rare variants (instead of common variants) in the cases. The real challenging is that very few people will have access to the full individual data in gnomAD to detect LD in rare variants. Therefore, we attempted to address this very challenge in the controls given no individual full genotype. On the other hand, if LD test is performed in the cases without knowing the actual phase, the compound heterozygous variants can be detected as significant pairs in LD, and cannot be distinguished from those rare variants in LD in the general population. For LD detection in the controls, we now added the comparison of LD detection using gnomAD and LD detection using available individual data (data from 1,000 Genomes Project and the larger 8,175 constructed control data) to identify unique contributions from analyzing gnomAD using our proposed method. There are indeed extra contributions from gnomAD (Fig. S7), and therefore we would like to keep this component. Indeed, we agree that the unique contribution from that using full genotype based controls is usually larger, which is also consistent with our power simulation results. Considering the two different data sources can complement each other, we now merged the LD detection results from both data sources and use it for checking LD under the recessive models. We summarized our analyses and added the following in the section: “The proposed LD detection method has high power in detecting high-LD rare variants and accurately identifies MNVs in gnomAD”

“We also compared the LD detected using gnomAD and LD detected using available individual data (data from the 1,000 Genomes Project and the larger 8,175 constructed control data). For individual genotype based data, we used the ld function from the R package snpStats. We focused on variants with gnomAD maximal allele frequency among ethnicities <0.1. LD test were performed within each ethnicity and the selection criteria of variants based on annotations were the same as that in the LD scan for gnomAD. We used FDR threshold 0.05 for each ethnicity. We stratified the significant LD pairs based on their alternate allele frequencies and calculated the contributions of each data source (Fig. S7). For relatively frequent variants, e.g., the AF range [0.01, 0.1), the unique contribution of gnomAD based LD detection is small (<3%), however, as the AF range becomes lower, the contribution of gnomAD based LD detection becomes substantial. For example, within the AF range [5e-4, 0.001), over 30% of the significant LD pairs can only be discovered using gnomAD based approach if compared with that using the 1,000 Genomes Project’s individual level data, the unique contribution is still about 25% if compared with that using the larger 8,175 constructed individual control data. These substantial contributions are likely due to the large sample size of different ethnicities in gnomAD. The unique contribution from gnomAD can be even higher when considering each ethnicity individually (Fig. S8). For example, in the AF range [5e-4, 0.001), the unique contribution of gnomAD based approach can be over 35% for the East Asian and South Asian, and about 80% for the Finnish population, even when compared with that using the larger 8,175 individual level

control data. On the other hand, the unique contribution from that using full genotypes is usually larger, which is also consistent with our power simulation results. Considering the two different data sources can complement each other, we merge the LD detection results from both data sources and use them for checking LD under the recessive models.”

3) When they excluded variants in LDs for the association analysis under the CoCoRV framework, it's unclear why those top associations are false positives (in Fig. 5). It's also unclear why all these association signals for variants in LD were removed? Should remain one association signal from the most possible causative variant?

Response:

Thanks for your comments. After checking the variants that drive the signal, we found that those top associations of individual genes under the recessive models are all due to high LD between two variants. These high LD variants are counted independently in the recessive model if LD among variants is not considered, but are counted only once in the recessive model when LD among variants is considered. We have added a table to show the exact variant pair in high LD that drives the association signals (Table S7). One example is the variant pair (3-37089130-A-G, 3-37089131-A-C) in gene MLH1, which shows high LD between the two variants, and is also identified as an MNV from gnomAD.

We added the following in the section: “Accounting for LD between two variants”

“When estimating the counts of qualified samples in controls for each group of rare variants in LD, only the variant with the highest AF was kept representing the variants in LD; the rest were excluded. The reason that we exclude redundant variant pairs that are in LD in the controls from the counting of the recessive model is twofold: 1) to match the assumption when estimating the number of samples under the recessive models in controls: rare variants are assumed to be independent; 2) it is less interesting if the recessive pattern in the case samples can be simply explained by LD between two variants in the control population. Fig. S15 illustrates the counting process under different models with/without considering LD between pairs of rare variants in LD.”

We added the following to the section: “CoCoRV analysis of the CNS and ALL cohorts”

“However, when the LD is accounted for in CoCoRV, there is no gene passing the exome wide significance. This suggests that for the CNS and ALL, the association signal is mainly driven by the dominant model, although it is also possible that we do not have enough power for the recessive models with the current sample size.”

4) In Method section of Estimation of sample counts in three models using summary genotype counts. For formula (2)-(4), the authors assumed independence between the rare variants. It's unclear how to calculate a burden test for a gene with multiple variants if they don't have individual data (no information to tell if a subject carry alternative alleles of multiple variants for a gene of interest). This could be a limitation to be discussed for the gene-based association

analysis.

Response:

Thanks for pointing this out. Given the summary count information for each individual variant, we use the formula (2)-(4) to estimate the probability of a sample falling into each of the three models assuming independence between rare variants. Then the **expected** total sample count of each model can be calculated as $n_{control} \times p_{model}$. For a burden test, we only need the total sample count (for individuals carrying the gene) under each model for cases and controls. For cases, we can directly count them; for controls, the **estimated expected** sample count in controls serve as an estimation of the actual total sample count if the individual genotypes in controls were given. The accuracy of the estimation relies on the rareness and independence of variants. For example, if the allele frequency of two variants is 10^{-3} and they are independent, the chance of observing both alternate alleles in the same haplotype is 10^{-6} , which is almost negligible compared to observing one alternate allele ($\sim 2 \times 10^{-3}$). In this case, ignoring the probability of multiple alternate alleles within one single sample can still be a good estimate for a dominant model. For the same reason, when considering the double heterozygous models, we only consider the probability of two variants with alternate alleles and ignoring the scenario of three or more variants with alternate alleles within a sample because those probabilities are much smaller. Simulations show that the estimation is quite close to the actual counts (Supplemental text, Fig. S9). Because the sample count for controls is still an estimation, we agree with the reviewer about the limitation and added the following in the discussion:

“For example, our proposed method estimates the sample counts in the controls under different models rather than counting the actual counts from individual level genotypes, there is possibility that the estimated counts might not match well with the actual counts. This is the limitation of using summary-count based burden tests.”

Reviewer #2 (Remarks to the Author):

The authors report a refinement and improvement in a previously published method to find genes implicated in rare diseases. More information is needed to understand how this updated method change is the scientific conclusions drawn from this type of analysis.

Response:

Thanks for the comments. Although we are not the first to leverage the large publicly available summary statistics from gnomAD or other control cohorts, our methods have quite a few novel features. During the revision, we have added more comparisons between our method and other related tools to illustrate the reduced false positives and increased power for novel gene discoveries. In addition, we have further improved the speed of the estimation of inflation factors by sampling directly from p-values' cumulative distribution function (CDF) under the null hypothesis of no association (revised Figure 2). The computing time is less than one minute for sampling 1,000 replications across $\sim 20,000$ genes. For FDR control, in addition to the powerful

resampling based FDR method, we also extend the FDR method designed directly for discrete and heterogeneous distributions from Fisher's exact test to the CMH exact test (revised Figure 2a, Section Inflation factor estimation for discrete count based test & FDR control). We now added a feature for association analysis contrasting functional and non-functional alleles as proposed in ProxECAT². For a clear comparison, we have added a table (Table S1) to summarize the features among different tools for summary count-based analysis. We believe our developed tool addresses several key problems when using large publicly available summary counts for gene prioritization, such as consistent variant filtering, ethnicity stratified analysis, inflation estimation, FDR control, and accounting for LD, in order to achieve reduced false positives and improved power. Besides the software that we developed, we believe our comprehensive analyses and comparisons of summary counts based analyses shed light on the advantages as well as limitations of using summary counts for rare variant association tests.

It would be helpful for each analysis to:

1) Annotate with allele frequency using neither the case nor the control data set. For example, the use of TOPMed could be considered.

Response:

Thanks for your suggestion.

First, we would like to clarify our allele frequency (AF) filtering strategy. In our AF based filtering, we neither use the case-derived AF nor the control-derived AF, but the joint AF estimated from the pooled cases and controls. This is the filtering strategy commonly used when full genotype case-control data is available such as in GWAS and does not introduce potential biases. However, previous summary count based tool such as TRAPD does not follow this strategy, instead, it uses AF only from the controls (gnomAD) for filtering. Filtering variants using only case-derived AF or control-derived AF, or filtering cases using case-derived AF and filtering controls using control-derived AF can introduce inconsistencies, biases, and inflated type I error.

In principle, AFs derived from an independent large non-rare disease cohort should be helpful and can also provide an unbiased way for AF based filtering. We followed the suggestion and added the results of using TOPMed Freeze 8 AF annotations for variant filtering (Table S8, rows in light blue). Unfortunately using TOPMed AF alone introduced false positives passing the exome-wide significant threshold ($p < 2.5e-6$). After checking the driving variants of those false positive genes, we found that the most important cause is the difference in allele frequencies between gnomAD V2.1 and TOPMed, which can be attributed to the following factors:

1. Ethnicity specific AFs. There are important differences in ethnicity proportions between gnomAD and TOPMed. One example is the variant 1-18809351-G-C, which has the largest AF in gnomAD V2 South Asian (0.1285) and an overall AF 0.01652 in gnomAD V2. TOPMed only provides the overall AF which is 0.00082360. This suggests that TOPMed likely has much lower proportion of South Asian samples than gnomAD.

2. Differences in gnomAD and TOPMed pipelines. While gnomAD uses the GATK pipeline, TOPMed uses its own GotCloud pipeline which uses samtools to generate genotype information (<https://genome.sph.umich.edu/wiki/GotCloud>). For some variants, the AF is relatively high in gnomAD but not available in TOPMed. One example is 20-44520259-CTG-C (gnomAD overall AF 0.01765), but not available in TOPMed. This variant also shows large differences among ethnicity specific AFs, 0.0758 in East Asian and 0.00084 in non-Finnish Europeans.

Because of the above differences, variants that should be filtered out, such as those with relatively high AF in a specific ethnicity, were kept when TOPMed's AF is used for filtering, resulting in the false positives. Nevertheless, TOPMed could still be used as an *additional* AF based filtering source to enhance our method. When applying both the joint AF based filtering and TOPMed based AF filtering in CoCoRV (CoCoRV_topmed_jointAF in Table S8), we don't see any false positives achieving exome wide significance.

We summarize the above analyses in the Discussion:

“In principle, allele frequencies derived from an independent large non-rare disease cohort should be helpful and can provide an unbiased way for AF based filtering. We tried to use TOPMed AF (Freeze 8) annotations for variant filtering. Unfortunately using TOPMed AF alone introduces false positives (Table S8). One main reason is that there are likely substantial differences in ethnicity proportions between gnomAD and TOPMed. Variants that show large ethnicity specific AFs and should be filtered out are kept if we only use TOPMed's AF for filtering. One example is the variant 1-18809351-G-C, which has the highest AF in gnomAD V2 South Asian (AF = 0.1285), but TOPMed's overall AF is very low (AF = 0.00082360). Even though using TOPMed AF annotation alone for AF filtering can cause false positives, TOPMed could serve as a useful resource for an additional AF based filtering on top of the joint AF based filtering.”

2) Uniformly report results when comparing between two methods. For example in this assessment X number of genes using Y program and the C number of genes using B program achieved exome wide significance. Of those that achieve this level of significance, Y program generated xx% of false positives and B program generated XX percent of false positives. These false positives were determined to be due to the following issues. I cannot tell in the results if the authors are reporting on false positives for signals that didn't cross an exome-wide significant threshold.

Response:

Thanks for your great suggestion. We have added Table S8 (the dominant model) and Table S10 (the recessive model) to summarize all the comparisons for four data sets we analyzed using different methods and configurations. For each method, we report the number of genes passing the exome-wide significance ($p < 2.5e-6$), the percentage of false positives, and the cause of these false positives as well as representative variants leading to the false positives. We hope that this side-by-side comparison significantly increases the clarity and readability of our manuscript.

We revised the paragraphs in the section “CoCoRV analysis of the CNS and ALL cohorts” as follows:

“... For TRAPD, using the option "--pass" shows less false positives than using the QD based filtering (Table S8), therefore we used the former for comparison with CoCoRV. For the CNS cohort, TRAPD reported an inflated $\lambda_{2points}$ (Fig. 6a). Two genes passed the exome wide significance ($p < 2.5 \times 10^{-6}$) using TRAPD. One was a false positive and the other was the known glioma causal gene NF1. For CoCoRV, it showed no inflation in either the pooled counts from all ethnicities (Fig. 6b) or the stratified analysis using CMH (Fig. 6c). One gene passed the exome wide significance, which was the known causal gene NF1. After manual variant checking, the false positive from TRAPD was caused by inconsistent filtering: the variant has FILTER status PASS in cases and therefore included in cases but has the failed status (failed the random forest based QC) in gnomAD therefore not included in controls (Table S8). For the ALL cohort, five genes passed exome wide significance using TRAPD and four of them were false positives. The inflation factor $\lambda_{2points}$ was inflated (Fig. 6d). In contrast, CoCoRV showed no obvious systematic inflation, and identified the known causal gene ETV6 as the only exome wide significant gene. The CMH-based analysis had better calibration at the tail than did the FET-based analysis (Fig. 6e, f). The cause of the false positives from TRAPD was either inconsistent filtering between cases and controls or low quality variants which showed large differences in AF between gnomAD WES and WGS data (Table S8).

...

We also explored the recessive models in the association test using both TRAPD and CoCoRV (Table S10). When the LD was not accounted for, for both TRAPD and CoCoRV, all genes that passed the exome wide significance were due to the high LD between variants, therefore considered as false positives. However, when the LD was accounted for in CoCoRV, there was no gene passing the exome wide significance. This suggests that for the CNS and ALL, the association signal is mainly driven by the dominant model. It is also possible that we do not have enough power for the recessive models with the current sample size of cases.”

3) there are several programs that allow for this type of rare variant burden testing, is there a reason why the authors only chose to compare to TRAPD?

Response:

Thanks for your comments. If full genotypes of both cases and controls are available, many programs have been implemented for rare variant burden tests, such as SKAT/SKATO in the SKAT R package. However, when only the summary counts of the controls are available, to our knowledge, TRAPD seems to be the only recent and well-designed program that provides an almost end-to-end pipeline for the rare variant burden analysis.

Through a more exhaustive search, we found another method ProxECAT² which pools the functional and non-functional alleles from cases and controls into a contingency table and then testing the enrichment of rare variants. However, this method does not deal with the upstream

QC steps like our method and TRAPD. Nevertheless, we implemented the ProxECAT method as one component in CoCoRV after the count matrices are obtained by our workflow. We also compared its performance on the same four data sets that we analyzed. We summarize the results as follows in the section: CoCoRV analysis of the CNS and ALL cohorts

“ProxECAT² is another recently proposed method for performing summary count based analysis. It pools the counts of functional and non-functional alleles from cases and controls to form a contingency table, which is used for an association test. We implemented this method within CoCoRV and applied it to the CNS and ALL cohort. ProxECAT showed slight inflations (inflation factor 1.05 for ALL, and 1.08 for LGG). The known gene NF1 and ETV6 reached the exome wide significance level and there were no false positives reaching the exome wide significance (Table S8). For the known causal genes NF1, CoCoRV shows more significant p-values ($p=7.6\times 10^{-10}$) than ProxECAT ($p=9.2\times 10^{-8}$), the trend is the same for ETV6 too (Table S9).”

We also added the following in the Method section: Comparison of CoCoRV with other methods

“Another method ProxECAT² which pools the counts of functional and non-functional alleles between cases and controls to form a 2×2 contingency table for the test of the enrichment of rare variants. However, the R package ProxECAT only provides two functions performing the statistical tests assuming the contingency tables have already been generated. There is no tool for any upstream filtering or generation of the 2×2 contingency tables. For a fair comparison, we implemented the pooling method proposed by ProxECAT in CoCoRV, and used the same upstream variant filtering from CoCoRV. The functional alleles were defined the same as that in CoCoRV for pathogenic alleles. The non-functional alleles included alleles annotated as synonymous or missense variants with REVEL <0.2 . The test in ProxECAT is essentially the likelihood ratio test given the 2×2 contingency table, which might not produce accurate p-values when the counts are very small, therefore we replace it with Fisher’s exact test in our analysis.”

If we still miss other methods that were designed for the same purpose, we are grateful if they can be shared with us. We will be happy to have a look.

4) The authors allow for consideration of ethnic specific minor allele frequencies —How much did this really change the results? Was this impact limited to African populations?

Response:

Thanks. This is indeed a very good question that prompted our investigation. We now added a section “The impact of ethnicity compositions on summary count-based analysis” to specifically to discuss this.

“Population structure is a known confounder for genetic association test if not properly addressed³. The effect of ethnic specific allele frequencies (AFs) on the association test results depends on the differences among ethnic specific AFs, and the differences in ancestry compositions between cases and controls. The majority population of gnomAD V2 exomes is European (nfe = 47%), and the four data set we analyzed (ALL, CNS, GBM, LGG) all have

European populations as the majority ($nfe \geq 70\%$, Fig. S13). The dominance of European samples in the cases and controls, as well as our focus on rare variants ($AF \leq 1e-3$), might explain that in general, the results using ethnic specific AFs or the pooled AFs did not show large differences in terms of genes achieving exome wide significance for the four data set we analyzed. However, we do observe one difference under the recessive model. Although LD was accounted for, using the pooled AF in analyzing the CNS cohort resulted in one false positive ($p\text{-value} < 2.5 \times 10^{-6}$), while there was no false positive if ethnicity stratified analysis was used (Table S10). The variant causing the false positive is 19-55898120-A-AT, which has the largest AF in gnomAD V2 African/African American ($AF = 0.1122$), but the AF is very low in European population ($AF = 0.0002196$).

To further illustrate the impact of ethnicity compositions and the importance of ethnic specific analysis, we constructed a simulated “case” cohort using samples from the 1,000 Genomes Project with sample ID starting with “NA”. In total, there were 840 samples included from the five major populations (nfe : 210, afr : 250, amr : 65, eas : 213, sas : 102). The ethnicity composition in the constructed “case” was very different from that in gnomAD, e.g., the European samples were not the majority. Because the samples from the 1,000 Genomes Project are not disease-specific, genes that were significant in the analysis between the constructed “case” and gnomAD were false positives due to the confounding of population structures, instead of being associated with any disease. Association results showed that the simple pooled analysis can produce many false positives (Fig. S14 a, d, Table S11), while ethnicity stratified analysis substantially reduced the false positives (Fig. S14 c,f). Adding the filter using the maximal allele frequencies among gnomAD ethnicities (AF_{popmax}) could also mitigate the false positives (Fig. S14 b,e), although it was not as effective as ethnicity stratified analyses. The variants that drive the top false positives in the simple pooled AF analysis showed large differences in allele frequencies between different ethnicities, where the AF ratios between the highest AF and lowest AF among ethnicities were around 10 and could reach about 20 (Table S12). The number of false positives decreases as the AF threshold decreases. For example, when AF was set to $1e-4$ and the ethnicity stratified analysis was used, there was no false positives in the analysis. The impact of different ethnicity compositions was not just limited to the African population but could happen in any other populations (e.g., eas , afr , amr , sas) less represented in the gnomAD data set (Table S12).”

5) TRAPD allows you to annotate and select variants based on a variety of metrics. Is there a reason why the authors are limiting it to a single metric?

Response:

Thanks for raising this question. We originally used the single metric QD when evaluating TRAPD is because it was recommended in the TRAPD paper. Now we have also added another comparison using the option `--pass` which filters the variants based on the FILTER field in the VCF file. Besides, we now used the option `"--minAN"` to make sure the variant missingness ≤ 0.1 , which can help reduce false positives. The option `"--popmaxAF"` is used for filtering

based on AF_popmax in gnomAD. Based on the new results, we revised Fig. S16 accordingly. We found that for TRAPD, using the "--pass" option indeed performs better than the QD based filtering in our analyzed data sets (Table S8), therefore we now used this configuration for the comparison under the recessive model (Table S10).

We added the following in the section: Comparison of CoCoRV with other methods

“... Besides the QD based filtering, we also tried the option "--pass" for variant filtering which is based on the FILTER status. Because the filtering is performed in cases and controls separately, it can create inconsistencies. We used the option "--minAN" to make sure the variant missingness ≤ 0.1 , which can help reduce false positives. The option "--popmaxAF" is used for filtering based on AF_popmax in gnomAD. We use TRAPD's "--includeinfo" to select variants based on the defined criteria of pathogenic variants.”

References

- 1 Clayton, D. & Leung, H. T. An R package for analysis of whole-genome association studies. *Hum Hered* **64**, 45-51, doi:10.1159/000101422 (2007).
- 2 Hendricks, A. E. *et al.* ProxECAT: Proxy External Controls Association Test. A new case-control gene region association test using allele frequencies from public controls. *PLoS Genet* **14**, e1007591, doi:10.1371/journal.pgen.1007591 (2018).
- 3 Price, A. L. *et al.* Principal components analysis corrects for stratification in genome-wide association studies. *Nat Genet* **38**, 904-909, doi:10.1038/ng1847 (2006).

Reviewers' Comments:

Reviewer #1:

Remarks to the Author:

Thank the authors for their excellent and responsive revision. I only have additional comments about their responses for my previous question 2a. In order to demonstrate the accuracy of LD estimation using the log likelihood of the observed data from six independent sets (i.e., stratified by Sex), it will be helpful if the authors can compare LDs estimated from summary counts with those based on individual-level data. They could randomly assign six groups from individual subjects to calculate summary counts based on their in-house individual-level data and then estimate LD using their developed log likelihood approach based on these summary counts (i.e., similar to the scenario of using summary counts from six groups using data from genomAD). By comparison, they can calculate LDs using `ld` function in `snpStats` based on their in-house individual-level data as they did in the revision.

Reviewer #2:

Remarks to the Author:

The authors have thoughtfully responded to all reviewer inquiries. They have improved their methodology. The revised paper provides a richer and clearer understanding of how powerful CoCoRV is in detecting disease-risk genes enriched with rare pathogenic variants. This represents a significant advance and is deserving of publication.

Response to reviewers

We thank both reviewers for their encouraging comments on our previous revision and have performed analysis to address additional comments below.

Reviewer #1 (Remarks to the Author):

Thank the authors for their excellent and responsive revision. I only have additional comments about their responses for my previous question 2a. In order to demonstrate the accuracy of LD estimation using the log likelihood of the observed data from six independent sets (i.e., stratified by Sex), it will be helpful if the authors can compare LDs estimated from summary counts with those based on individual-level data. They could randomly assign six groups from individual subjects to calculate summary counts based on their in-house individual-level data and then estimate LD using their developed log likelihood approach based on these summary counts (i.e., similar to the scenario of using summary counts from six groups using data from gnomAD). By comparison, they can calculate LDs using *ld* function in *snpStats* based on their in-house individual-level data as they did in the revision.

Response: To further show the accuracy of rare variant LD estimation, as suggested, we now added additional comparisons between LD test using summary counts from randomly assigned six groups and that using the individual-level data. The summary counts based LD test achieves high precision, consistent with that using gnomAD summary counts for LD detection. The relative power is about 15% compared with that using individual-level data, consistent with our power simulation studies and suggesting the advantage to use full genotypes when they are available. We have added the following in the manuscript:

“To further demonstrate the accuracy of our proposed LD detection method, we compared the LD detection results using either pooled summary counts or individual genotypes from our constructed in-house 8,175 synthetic controls. Specifically, in each ethnicity, we randomly split the control data set into three data sets with sample size ratio 1:2:3, then we stratified each of the three data set by sex, generating six independent data sets. For each of the six data set, we pooled the alternate allele counts and the total allele counts, similar as that from gnomAD, and tested LD between variant pairs using CoCoRV. For individual genotypes, we used the *ld* function in the R package *snpStats*¹³ to test each LD pairs within each ethnicity. We focused on variants with gnomAD maximal allele frequency among ethnicities <0.1 and the selection criteria of variants based on annotations were the same as that in the LD scan for gnomAD. We consider 45,626 LD pairs detected using individual genotypes ($FDR < 0.05$) as the ground truth. In total, 6,694 LD pairs were detected ($FDR < 0.05$) using summary counts, where 8 were false positives, indicating high precision (99.8%) of CoCoRV. The lower power using summary counts (14.65%) is consistent with our power simulations, suggesting the power advantage of using full genotypes when they are available. For the true positives detected by CoCoRV, the estimated LD measure r^2 between using summary counts and using full genotypes were highly correlated (correlation = 0.853) (Fig. S7a). After stratifying detected LD pairs by alternate allele frequencies within the controls and the estimated LD measure r^2 from using full genotypes (Fig.

S7b), we observed that summary counts based LD detection mainly detect those high LD pairs with decent alternate allele counts (e.g., > 10).”

Reviewers' Comments:

Reviewer #1:

Remarks to the Author:

The authors have adequately addressed my concerns.